

# Mediterranean Specific Climate Classification and Future Evolution Under RCP Scenarios

Antoine ALLAM[1,2], Roger MOUSSA[2], Wajdi NAJEM[1], and Claude BOCQUILLON[1]

[1]CREEN, Université Saint-Joseph, Beirut, Lebanon
[2]LISAH, Univ. Montpellier, INRA, IRD, SupAgro, Montpellier, France

**Correspondence:** Antoine ALLAM (antoine_allam@hotmail.com)

**Abstract.** The Mediterranean is one of the most sensitive regions to anthropogenic and climatic changes mostly affecting its water resources and related practices. With multiple studies raising serious concerns of climate shifts and aridity expansion in the region, this one aims to establish a new high-resolution classification for hydrology purposes based on Mediterranean specific climate indices. This classification is useful in following up hydrological, (water resources management, floods, droughts, etc.), and ecohydrological applications such as Mediterranean agriculture like olive cultivation and other environmental practices. The proposed approach includes the use of classic climatic indices and the definition of new climatic indices mainly precipitation seasonality index $I_s$ or evapotranspiration threshold $S_{PET}$ both in line with river flow regimes, a Principal Component Analysis to reduce the number of indices, K-Means classification to distribute them into classes and finally the construction of a decision tree based on the distances to classes kernels to reproduce the classification without having to repeat the whole process. The classification was set and validated by WorldClim-2 at 1-km high resolution gridded data for the 1970-2000 baseline period and 144 stations data over 30 to 120 years, both at monthly time steps. Climatic classes coincided with a geographical distribution in the Mediterranean ranging from the most seasonal and dry class in the south to the least seasonal and most humid class in the North, showing up the climatic continuity from one place to another and enhancing the visibility of change trends. The MED-CORDEX ALADIN historical and projected data at 12-km resolution simulated under RCP 4.5 and 8.5 scenarios for the 2070-2100 period served to assess the climate change impact on this classification by superimposing the projected changes on the baseline high resolution classification. Both RCP scenarios showed a 7% to 9% increase of the average seasonality index $I_s$ and 3% to 20% increase of the average aridity index $I_{Arid}$ for the least seasonal classes. These classes located to the north are slowly evolving towards moderate coastal classes which might affect hydrologic regimes due to shorter humid seasons and earlier snowmelts. This kind of classification might be reproduced at the global scale, using same or other climatic indices specific for each region highlighting their physiographic characteristics and hydrological response.

## 1 Introduction

Mediterranean climate is a result of a complicated cyclonic system swiping a large evaporative basin. The distribution of marine and continental air masses creates an alternation of low-pressure zones coming over from Iceland and the Persian Gulf or high-pressure zones from Siberia and Azores. The seasonal shifts of these zones are magnified by the North Atlantic Oscilla-





tion (NAO) that plays an important role in shaping Mediterranean climate and influencing the evolution of farming and social activities on the long-term (Rodwell and Hoskins, 1996). This continuous alternation of high and low pressure, cold and humid winters followed by a hot and dry summers, marks Mediterranean seasonality which makes the region attractive to social activities, thus its sensitivity to climate change and anthropogenic pressures (Planbleu, 2012). Climate change and anthropogenic
pressures are expected to have severe consequences on Mediterranean runoff with a serious risk of water shortages (Hreiche et al., 2007; Cudennec et al., 2007; García-Ruiz et al., 2011; Verdier and Viollet, 2015). Hydrologically, this seasonality plays a role in shaping rivers runoff as Haines classified the Mediterranean under Class 12 Winter Moderate hydrologic regimes, Class 13 Extreme Winter and Class 14 Early Spring, and found a clear relation to the Köppen Csa and Csb climates and a close equivalent of the 'Mediterranean Seasonal' categories of Gentilli (Haines et al., 1988). Seasonality is a main factor in the
Mediterranean but to our knowledge its use is still limited as a characterising index for climatic and hydrological classification.

     This study is a contribution to the HyMeX program and to the Med-CORDEX initiative. The HyMeX progam (HYdrological cycle in the Mediterranean Experiment) aims at a better understanding of the Mediterranean hydrology, with emphasis on the predictability and evolution of decadal variability in the context of global change. MED-CORDEX is part of the COordinated Regional Downscaling EXperiment specific for the Mediterranean that aims at improving our understanding of climate change
through high resolution Regional Climate System Models (RCSM). RCP or Radiative Concentration Pathway is a greenhouse gas (GHG) concentration trajectory adopted by the International Panel for Climate Change (IPCC) for its fifth Assessment Report (AR5) in 2014. RCP 4.5 and 8.5 were chosen between 4 available scenarios being the most focused on in literature. RCP 4.5 assumes that global annual emissions measured in $CO_2$-equivalents peak around 2040, with emissions declining substantially thereafter while under RCP 8.5 emissions continue to rise throughout the 21st century. The RCP 4.5 (resp. RCP
8.5) means that the GHG and aerosols concentrations evolve in a way that leads to an additional radiative forcing equal to +4.5 W/$m^2$ (resp. + 8.5 W/$m^2$) at the end of the 21st century with respect to the pre-industrial climate. Consequently, the RCP 4.5 can be considered as an optimist scenario whereas RCP 8.5 is a more pessimist option. (Giorgi et al., 2009; IPCC, 2013; Ruti et al., 2016).

     While temperature increase and precipitation decrease have been already observed (IPCC, 2013), MED-CORDEX RCP 4.5
scenario projections, as simulated by ALADIN v5.2 for the 2071–2100 period (Tramblay et al., 2013), estimates a spatially distributed temperature increase of 1.4 to 3.5°C and a precipitation evolution of ± 10% while RCP 4.5 projects an increase of 2.2 to 6.4°C and a precipitation evolution of ± 20% compared with the baseline period 1970-2000 with expected shifts of Mediterranean climate and expansion of arid regions (Beck et al., 2018; Barredo et al., 2019) and related water restrictions and legal decision-making processes (Sauquet et al., 2018).

There are several climate classification studies of the Mediterranean; among these we start with Köppen-Geiger classification at the global scale in which the Mediterranean climate is well distinctive (Köppen, 1936; Peel et al., 2007) and the decomposition into climatic regions at the national scale in France, Turkey and other (Erinç, 1984; Champeaux and Tamburini, 1996; Unal et al., 2003; Sönmez and Kömüşcü, 2011; Eveno et al., 2016). We also mention the classification of synoptic meteorology (Trigo et al., 1999; Alpert et al., 2004), cloud classification (Chéruy and Aires, 2009), and the hydrological classification of





natural flow regimes, (Belmar et al., 2011; Oueslati et al., 2015). However, no specific classification based on precipitation and temperature series has yet treated the Mediterranean region as a climatic or hydrological unit, hence the aim of our study.

The objective of this study is first to establish a Mediterranean specific climatic classification for hydrology purposes based on a set of indices mainly seasonality, second, to estimate future evolution of this classification based on RCP scenarios with an easy follow up visual tool and third, to assess the utility of this classification in a coupled physiographic-climatic environment illustrated by olive cultivation evolution in the Mediterranean.

Three types of climatic data were used for this purpose, (1) WorldClim-2 new 1-km spatial resolution climate surface data (Fick and Hijmans, 2017), (2) Monthly average time series of 144 stations from NOAA database and cover 20 different Mediterranean countries for a period of 30 to 120 years used for validation purpose (3) MED-CORDEX historical and projected data at 12 km resolution simulated under RCP 4.5 and 8.5 scenarios for future projections. (Tramblay et al., 2013).

The suggested methodology includes first the definition of the climatic indices from which some are classic like the frequency indices and other are specific of the Mediterranean climate like the seasonality. Second, a Principal Component Analysis (PCA) to reduce the number of climate indices and consider only the most contributing. Third, K-Means classification according to the most contributing indices and finally the construction of a Decision Tree based on distances to classes kernels to determine whether or not a place has a Mediterranean climate, and to which type it belongs. This approach was applied at catchment scale, where climatic indices are averaged for each catchment and then validated at grid scale and a set of ground stations. Each class was described and characterized by its corresponding climatic indices. The Mediterranean climatic classes evolution was assessed according to indices variation based on simulated RCP scenarios and expressed by olive tree cultivation as an example of historical Mediterranean specific bioindicator. Also, olive reproductive cycle displays considerable variations due to climate evolution among other, influencing flowering intensity mainly affected by seasonal temperature and water availability (Moreno, 2014).

This paper is structured into six sections; Section 1 Introduction; Section 2 presents the database and the Mediterranean limits; Section 3 the classification approach based on PCA, K-Means and the decision tree; Section 4 the results of WorldClim-2 classification of catchment indices and validation on gridded indices and stations and Section 5 classification projection and impacts under MED-CORDEX RCP 4.5 and 8.5 scenarios before concluding in Section 6.

## 2   Study area and Database

### 2.1   Defining the Mediterranean region boundaries

From the Latin word Mediterrănĕus meaning 'middle land' the Mediterranean refers to the sea and bordering region located in the middle of the Ecumene between the European, African and Asiatic continents. With Köppen's classification (Köppen, 1936) the definition designated henceforth a moderate climate and extended geographically beyond the limits of the Mediterranean Sea. The question that arises is how would the Mediterranean boundary be defined? Several alternatives are considered in this case based on the practiced discipline, but topographical boundary was adopted for this study as shown in Figure 1.





- The climatic boundary could be defined according to Köppen's classification where a set of regions share similar temperature and precipitation characteristics and known for their warm and dry summers and cold and humid winters. It is limited by the African desert to the South and the temperate European countries to the North. This boundary might change according to the definition of this similarity. Some regions share a similar Mediterranean climate although located far outside the Ecumene

such as Chile, California or South Africa.

- The topographic boundary is defined by the set of catchments draining towards the Mediterranean Sea (Milano, 2013). This definition neglects some of Mediterranean climate regions like Portugal, part of Spain and favours geographically adjacent regions like Egypt and Libya.

- The agricultural-bioclimatic boundary consists of the set of regions sharing the same types of vegetation considered as
indicators of the Mediterranean region such as olives, (Moreno, 2014). This definition is linked to human activity with the same nuances as the climatic limit.

- The administrative boundary of countries adjacent to Mediterranean Sea has a problematic definition independent of any natural base (Wainwright and Thornes, 2004). These boundaries include several climatic classes and cover larger areas than the topographical limits.

## 2.2    Catchments

Since the geographic extent of the study is very wide to be treated in a personal way, the delimitation of catchments was imported from international references. The European Commission using the Joint Research Centre (JRC) has done extensive and elaborate work on the delimitation of catchments in Europe and some adjacent countries as part of the "Catchment Characterization and Modelling" (CMM) project (De Jager and Vogt, 2010). For catchments in the Middle East and Northern
Africa, catchments from HydroSHEDS, the World Wildlife Fund's project, were used (Lehner and Grill, 2013). According to these databases, the total number of catchments exceeding 1 $km^2$ and having a Mediterranean sea mouth outlet is 3681 covering a total area of 1,781,645 $km^2$. It should be noted that the Nile was omitted for its extent 3500 km to the south of the Mediterranean. Catchments surface distribution is shown in Table 1 where middle range catchments, between 100 and 3000 $km^2$, constitute 35% of the total and cover 28% of the total area.

## 2.3    Climatic data

Three climatic datasets were used for this study,

(1) WorldClim-2 new 1-km spatial resolution climate surface data, which consists of long-term average monthly temperature and precipitation, solar radiation, vapor pressure and wind speed data, aggregated across a target temporal range of 1970–2000, using data from 9000 to 60000 weather stations (Fick and Hijmans, 2017). Worldclim-2 database is a refined and expanded
version of the 2005 "WorldClim-1 database" (Hijmans et al., 2005). This database covers the whole study area, thus climatic classification of Mediterranean catchments was possible. Monthly precipitation and temperature were averaged for each catchment and then climatic indices calculated at both catchment and grid scale. Both classifications were compared for validation. Climatic characteristics of Mediterranean catchment are summarised and illustrated in Table 2 and Figure 2, reflecting the wide





variability of mean annual precipitation ranging between 5 and 3000 mm and mean annual temperature ranging between -14 and +26°C where some catchments receive 50 times more than others the amount of precipitation while being 4 times colder.

(2) 144 ground weather station data covering the whole study area served to validate the Mediterranean climate classification with 105 stations located within catchments boundary and 39 outside. Also, 102 of these stations located within Köppen's (Csa)

and (Csb) Mediterranean climate and 42 outside. These stations are recognized by the World Meteorological Organization (WMO) and available for free access on the portal of the National Administration of Oceans and Atmosphere of the United States (NOAA). The length of data series ranges between 30 and 120 years at monthly time step.

(3) The MEDCORDEX climate projection, computed with the RCM ALADIN-Climate v5.2 at 12 km spatial resolution grid, was used to analyse the climate change impacts on the climatic classification for the end of the century projection period

2070-2100, and for two different Radiative Concentration Pathway scenarios (RCP 4.5 and 8.5) in comparison to the historical 1970-2000 baseline period which was also adopted from ALADIN historical run (Tramblay et al., 2013).

## 3   Methodology

Taxonomy aims to separate a population into several groups of similar characters. It was mainly developed by naturalists (Linnaeus, 1748). But Thornthwaite pointed out that climate classification does not follow the same approach since one goes

from one climate to another continuously, whereas the various species of fish, for example, are all different, in fact individualized (Thornthwaite, 1948). This continuity can be demonstrated using a fine intra-climate classification. To achieve this fine classification, it is essential to introduce measurable indices ensuring continuous variable scale.

Automatic classification methods partition a set of objects knowing their distances by pairs in a way to keep the classes as much homogeneous as possible while remaining distinct from each other. Like any classification, the adopted method depends

from the objective and its author. There are several modes of climatic classification: (a) Genetic classifications related to meteorological causes and the origin of air masses (Bergeron, 1928; Barry and Chorley, 2009). (b) Bioclimatic classifications based on the interrelation between vegetation type and climate (Holdridge, 1947; Mather and Yoshioka, 1968; Harrison et al., 2010). (c) Agro-climatic method based on the assessment of the Rainfall - Evapotranspiration balance for the estimation of agricultural productivity (Thornthwaite, 1948). (d) Climatic methods based on precipitation and temperature indices similarly

to the classification of Köppen in 1936 (Köppen, 1936) updated by Peel in 2007 (Peel et al., 2007) and which remains the most used, this method divided the globe into thirty climate zones and was based on a hierarchical approach. The Mediterranean climate corresponds to dry hot or dry warm summer where either the precipitation in the driest month in summer is below 40 mm or below the third of the precipitation in the wettest month in winter (Cs) and the air temperature of the warmest month is above 22 (Csa) or the number of months with air temperature above 10 °C exceeds 4 (Csb).

The (Cs) climate doesn't reign all over the Mediterranean region, some exceptions could be observed. A Desertic climate (BWh) dominates Egypt and Libya, (Bsk) Southeast Spain and (Cf) the regions of Thessaloniki and Veneto. On the other hand, and at a global scale, (Cs) climate is present in California, Chile and South Africa, Figure 1.





### 3.1 Principle Component Analysis

Principal Component Analysis (PCA) is widely applied to reduce the dimensionality of datasets and keeping the most representing and uncorrelated variables. This section presents a brief description of the method along with some of their applications in hydrology. For an extensive mathematical description and demonstration of these methods we advise to consult;

Krzanowski's Principles of multivariate analysis: a user's perspective (Krzanowski, 1988) Jollife's book Principal Component Analysis including a wide range of applications (Jolliffe, 2002).

PCA was first introduced by Karl Pearson (Pearson, 1901) and then developed by Harold Hotelling (Hotelling, 1933). Hotelling's motivation is that there may be a smaller *fundamental set of independent variables which determine the values* and conserve the maximum amount of information of the original variables (Jolliffe, 2002). This is achieved by transforming a

10 vector of $p$ random variables to a new set of variables, named Principal Components (PC), by looking for a linear function of the elements having maximum variance. And next looking for another linear function uncorrelated with the first and having maximum variance and so on up to $p$ PCs. It is hoped in general, that most of the variation will be accounted for by $m$ PCs, where $m < p$.

### 3.2 K-Means clustering technique

Cluster analysis consists of data points partitioning into isolated groups while minimizing the distance between same cluster data points and maximizing it between different clusters. One of the most popular clustering methods is the K-Means method introduced by Edward Forgy (Forgy, 1965) and MacQueen (MacQueen, 1967). It aims to minimize the square error objective function for distance optimization. The optimization steps begin with (1) kernels initialization, the kernel being a virtual point representing the statistical centre of a class, (2) updating classes, (3) re-evaluation of kernels and (4) repetition of steps

(2) and (3) until stabilization. The quality of the solution thus found strongly depends on the initial kernels. In its turn, kernel initialization is sensitive to the data dimensionality. The application of K-Means requires setting a number of classes, otherwise the optimization leads to as many classes as individuals.

K-Means gained in reputation the last decades and was widely applied in hydrology field for clouds classification from satellite imagery (Desbois et al., 1982), for climatic classification using measured and simulated timeseries (Moron et al.,

2008; Carvalho et al., 2016) for catchment classification based on streamflow characterization and precipitation (Toth, 2013). K-Means classification was applied, and catchments were distributed based on their distances to 5 classes kernels, for their geographical suitability, to determine whether they belong, or not, to a Mediterranean climate and to which type they belong to, if so.

### 3.3 Decision Tree

The purpose of a decision tree analysis is to classify a population into groups by predicting values of a dependent variable based on values of predictor variables. This procedure provides validation tools for exploratory and confirmatory classification analysis. In our case, the dependent variables are the climatic classes obtained from K-Means clustering while the predictor





variables are the distances to each clusters' kernels. This procedure was done for both catchments and gridded classification. The decision tree generates a set of classification rules usually used to classify new stations based on their distances to classes kernels. In this study, these rules were used in section 5 to classify RCP 4.5 and 8.5 projected indices. In this way, we have fixed the classes kernels indices of the 1970-2000 baseline period and calculated the distances of the 2070-2100 projected grid

to baseline to compare both the classification indices and spatial evolution. The decision tree might have differed if another kernel was forced into the first node, but kernel 1 was adopted as it yielded the highest accuracy rate.

## 3.4 Adopted methodology

The proposed methodology consists on calculating the climatic indices using WorldClim-2 monthly gridded data averaged at the catchment scale using ArcGIS zonal statistics. The climatic indices were then PCA-reduced and classified using K-

10 Means clustering. The classification was validated on WorldClim-2 gridded indices and ground stations indices. In addition to a decision tree built to classify projected indices and to avoid repeating the whole process. All PCA, K-Means and the decision tree where calculated using SPSS® software.

For climate change assessment and for better comparison, temperature and precipitation delta change were calculated between MED-CORDEX RCM ALADIN grids for the baseline 1970-2000 and projected 2070-2100 periods for RCP 4.5 and

15 8.5 and then superimposed to the WorldClim-2 grid through proximity analysis and spatial join. The indices of projected grids were then re-classified using the decision tree and compared to the baseline grid.

## 4  WorldClim-2 classification results and validation on gridded and station indices

This section details the climatic indices derived from the collected database, the results of PCA/K-Means classification of each set of indices and their validation on gridded and station indices with a decision tree for replicating the classification on new

stations or grids.

### 4.1  Hydrology driven climatic indices

The hydrology driven independent climatic indices were subjectively developed from WorldClim-2 monthly average data and divided onto four groups to highlight the Mediterranean seasonality hypothesis of the climate and its corresponding hydrological response. While the flow seasonality is clearly affected by the precipitation seasonality, the other indices help in fine tuning

this theory like monthly temperature and potential evapotranspiration variation. A complete list of indices with a description of each is in Table 3.

Group I: indices based on monthly precipitation from which we mention seasonality index $I_s$, peak indices $S_{P1.5}$, $S_{P2}$ and frequency indices $P_{25\%}$, $P_{75\%}$. $I_s$ is directly linked to Mediterranean flow regimes for expressing the precipitation ratio between the 3 most humid months and the 3 most dry months with values ranging from 0 to 1 (Hreiche, 2003). $I_s$ values

tending towards 0 express uniform distribution of precipitation along the year with a hydrological response lacking flood and





drought seasons while $I_s$ values tending towards 1 correspond to a normal distribution of precipitation with a hydrological response more likely to show flood and drought seasons.

Group II: indices based on monthly temperature expressed by the temperature lag between the coldest and warmest months $\Delta T_1$, frequency indices $T_{25\%}$, the number of months exceeding the average Mediterranean temperature $S_{Tm}$.

Group III: indices based on both temperature and precipitation expressed by $I_{Decal}$ the time lag between the coldest and most humid month.

Group IV: indices based on precipitation and evapotranspiration expressed by aridity index $I_{Arid}$.

## 4.2 PCA Results

The number of indices was reduced the first time based on the correlation matrix and the second based on PCA results. We
eliminated the strongly correlated indices (correlation higher than 0.85) and 11 indices were kept upon the first step.

- $I_s$ and $P_{75\%}$ are strongly inversely correlated (-0.959). $I_s$ was kept.
- $\Delta T_1$ and $\Delta T_2$ are strongly correlated (0.989). $\Delta T_1$ was kept.
- $T_{25\%}$, $T_{75\%}$ and $D_j$ are strongly inversely correlated (-0.954 and 0.874). $T_{25\%}$ was kept.
- $P_{25\%}$ and $I_{Hor}$ are strongly correlated (0.858). $P_{25\%}$ was kept.

Once the correlation matrix transformed into a diagonal one, it was possible to find the eigenvalues representing the projection from p to k dimensions. The eigenvector matrix is the linear expression of the indices with respect to the principal components. The first eigenvalue 6.36 represents 58% of the variability and the second 1.31 represents 12%. The first two factors F1 and F2 represent the two greatest variabilities with respect to the following factors and 70% of the total variability is thus preserved with this choice. Upon the PCA, the number of indices was reduced to 7 showing that $I_s$, $P_{25\%}$, $S_{P1.5}$,
$I_{Arid}$, $T_{25\%}$, $S_{PET}$ and $S_{Tm}$ were the most contributing climatic indices with 70% of total variance explained for the first two components. Statistical summaries are shown in Table 4 with $I_s$ values ranging between 0.2 and 1 with an average of 0.8 highlighting Mediterranean seasonality.

## 4.3 Climatic Classification of WorldClim-2 catchments indices

The K-Means classification showed in Figure 4 that a distribution into 5 classes was the most suitable for a geographical
representation of the Mediterranean catchments where

- Class 1: present in Egypt and Libya only, highlighting a desertic influence with few rain episodes registered per year, if any, expressed by $I_s = 0.99$ and $I_{Arid} = 11.7$ on average. Precipitation never exceeds evapotranspiration in this region, hence $S_{PET} = 0$.
- Class 2: mainly present in the south and east of the Mediterranean, characterised by a high seasonality $I_s = 0.95$ and high
aridity $I_{Arid} = 4.3$.
- Class 3: dominates the central region from Spain to Syria with an average seasonality $I_s = 0.91$.
- Class 4: covers the coastal catchments in north-western countries, south-east Italy, western Greece and present discontinuously in the south-west. $I_s = 0.71$ in this class.





- Class 5: only present in northern non-coastal catchments and characterised by a low seasonality $I_s = 0.47$.

In comparison to Köppen's, classes 2, 3 and 4 matches perfectly with (Csa) while classes 1 and 5 are mainly outside (Csa) and (Csb) henceforth defined as Non-Mediterranean climate. The main difference with Köppen's Mediterranean classes resides in Southern Spain defined as arid climate (Bsk) while in the present classification it varies between classes 2, 3 and 4. This

new distribution indicates climate variability within (Csa) or (Csb), hence the importance of a fine gridded classification. This variability is highlighted in the class kernels indices (Figure 3). It is mainly due to the complex seasonality across the Mediterranean. This complexity is shown here more delicately than the one defined by Köppen which is climate oriented only and limited to the simple criteria of a wet winter and dry or temperate summer. Therefore, we think that a hydrology oriented climatic classification should account for and intra climate characteristics expressed by specific indices like the one shown

here, specific to the Mediterranean and expressed by $I_s$. The continuous evolution of climate across the Mediterranean was demonstrated by the indices values uniformly increasing or decreasing from North to South. Seasonality is highest in the South and lowest in North, same for other precipitation indices and aridity.

## 4.4 Validation of the climatic classification

### 4.4.1 Validation for WorldClim-2 gridded indices

The K-Means clustering of WoldClim-2 gridded data resulted with a similar spatial distribution where class 1 dominates the south, class 5 in the north and classes 2, 3 and 4 in the middle (Figure 5). This classification has shown better resolution due to catchments averaging approximation and revealed some shifts to adjacent classes. Class 4 climate appeared on Spanish coasts, class 3 climate appeared on Sardinia and Greece, Class 2 in Syria and a limited spread of class 4 and 5 on Eastern Turkey. However, climate continuity is conserved in this classification for indices are gradually increasing or decreasing from North to

South.

Motivated by the quest for coupled physiographic-climatic models, we believe that this classification is useful both for hydrological and ecohydrological applications like cultivation and other related environmental practices affected by water resources and river flows. Olive is one of the best Mediterranean-specific physiographic indices and we noticed that its cultivation boundary is limited by those of classes 1 and 5 where 13% is in Class 2, 49% in class 3 and 34% in class 4. This observation

gives an accurate idea of perfect climate conditions for olive cultivation, deducing that extreme seasonality combined with very high aridity (South) or very low seasonality combined with high humidity (North) are avoided by olive trees. In a similar way, other tree types like pine trees also characterise Mediterranean landscape putting forward the need for a physiographic classification to interpret in parallel to this climatic classification under the umbrella of hydrological characterisation. The future of Mediterranean cultivation in case of climate change is to be checked under RCP 4.5 and 8.5 scenarios in next section.

### 4.4.2 Validation for stations indices

The 144 stations were also K-Means clustered based on the selected indices from the PCA. The resulting geographical distribution differed only by some shifting due to averaging and normalization as the sample is much less than the gridded cells.





There is no coverage of class 1 as no weather station was found in that region (Figure 6). Despite the shifting, there is an 82% accuracy rate or 86 out of 105 stations located within catchments boundary that matched the gridded distribution, the rest is located within the adjacent classes boundaries. As for olive boundary, there was only one class 5 station corresponding to Firenze that was located within the boundary.

### 4.4.3   Decision tree analysis validation

The total population of gridded classification was divided into two equal subsets, one for training and the second for testing. The predicted classes values of both sets were then compared to the original classification and both yielded an overall 93% accuracy for gridded classification (Table 5). We notice that some grids have joined one of the adjacent classes due to interclass connectivity; this confirms once more the continuity of climate. The generated decision tree of 3 levels includes 75 nodes in total due to high population number with 75 classification rules sampled in (Table 6). As an example, for class 1, if the distance to kernel 1 (D1) is below 3.5 and the distance to kernel 2 (D2) is above 2.2, then the grid cell belongs to class 1.

## 5   RCP 4.5 and 8.5 Scenarios Climate Evolution

For climate change impact assessment, temperature and precipitation delta change were calculated between both baseline period 1970-2000 and projected period 2070-2100 for MED-CORDEX RCM ALADIN grids and for two different Radiative Concentration Pathway scenarios (RCP 4.5 and RCP 8.5). Those delta changes were then transposed to the WorldClim-2 grid through proximity analysis and spatial join. The decision tree rules from Table 6 were then applied for the projected period and the climate change under RCP was illustrated in Figure 7 and expressed by indices evolution between classes in Table 7.

Under RCP 4.5 scenario, Mediterranean region temperature is increasing by 1.4 to 3.5°C and precipitation is decreasing by 10% in one third of the region but increasing by 10% in two third of it. Overall, the Mediterranean is evolving towards a moderate/arid region under RCP 4.5 scenario as no major area change has occurred but instead, classes 4 and 5 seasonality index $I_s$ is increasing by 7% and 9% while classes 1, 2 and 3 are constant. Also for classes 4 and 5, $S_{P1.5}$ is highly increasing (70%) with $P_{25\%}$ almost the same (2-3%) which means that the precipitation change that has occurred was temporally distributed in a way that more months are exceeding the average monthly precipitation by 1.5, so the humid season has become shorter enhancing seasonality variation. Another remarkable change is class 5 $I_{Arid}$ 20% increase pushing it towards class 4. In detail, even though no major area change was observed, classes distribution has changed where class 5 has reduced its extent in Greece and Albania in favour of classes 3 and 4 but compensated by the appearance of class 5 in central Spain. Class 3 extent has decreased in Turkey and Corsica in favour of class 4 in Lebanon and class 2 in Cyprus.

The case is similar but accentuated under RCP 8.5 scenario for temperature is increasing by 2.5 to 5.6°C and precipitation is decreasing by up to 20% in half the region and increasing by up to 25% in the other half. The difference with RCP 4.5 scenario resides first in the indices evolution where $I_s$ is increasing by 9% in class 5 but $S_{P1.5}$ is highly increasing by 96%. This change has caused an area change of 2% towards class 4 mainly in Spain, Greece and Albania. Another change occurred in class 3 where $I_{Arid}$ has increased by 19% and $S_{PET}$ decreased by 10% which means that this moderate region is pushing towards





more arid climate. In the spatial distribution details, although its area has not changed, class 3 is taking over the south eastern coast of Spain but retreating in favour of class 4 in North West Africa and Turkey.

The evaluation of uncertainties in mean precipitation over the 1981–2010 period,Colmet-Daage et al. (2018), found that the total monthly precipitation in spring and summer is overestimated over the mountainous regions and underestimated over the

5 coastal region. The mean and spread for future period remain unchanged under RCP4.5 scenario and decrease under RCP8.5 scenario. However, precipitation events are infrequent during spring and summer seasons in the Mediterranean except for Class 5 region which is characterized by the lowest seasonality.

The RCP 4.5 and 8.5 scenarios might look Mediterranean friendly as classes 4 and 5 seasonality indices are evolving towards class 3 in addition to some spatial expansion which make it more favourable for Mediterranean cultivation but not as much for

Mediterranean hydrology and water resources management, as temperature increase might affect snowmelt runoff discharge and consequently the hydrological regimes as per Haines classification mainly class 13 Extreme Winter taking over class 14 Early spring. The RCP 8.5 impact on hydrology is even more accentuated with less available resources caused by lower precipitation.

## 6 Conclusions

The Mediterranean climate characteristics and specifically precipitation seasonality, main contributor according to PCA, plays an important role in the hydrological mechanisms of Mediterranean catchments and flow intermittence. A decision tree makes it possible to define, from distances to class kernels, if any place has a Mediterranean climate or not, and to which type of Mediterranean climate does it belong to, for present and future scenarios. The interclass connectivity showed that climate is continuous from one place to another since some catchments or grid cells can meet the membership criteria of adjacent

classes. On the other hand, the superposition of olive cultivation boundary as Mediterranean-specific physiographic index highlighted the utility and importance of physiographic-climatic coupled scenario models that could be extended to other Mediterranean physiographic or bio-climatic indices. The climatic classification and corresponding indices evolution under RCP scenarios helped in identifying the general climate change impact on Mediterranean seasonality that might uncover valuable findings about water balance, floods and droughts for water sector stakeholders. Both RCP 4.5 and 8.5 scenarios

showed an increase of the average seasonality and aridity indices affecting hydrologic regimes due to shorter humid seasons and earlier snowmelts. The results of this study are useful for future water resources and cultivation management policies to identify the most impacted zones and propose preventive and adaptative measures for a more resilient region. This kind of classification might be reproduced at the global scale, using same or other region-specific climatic indices highlighting their physiographic characteristics and hydrological response.





*Acknowledgements.* This work is a contribution to the HyMeX program (HYdrological cycle in the Mediterranean EXperiment) through INSU-MISTRALS support and to the Med-CORDEX initiative (COordinated Regional climate Downscaling EXperiment – Mediterranean region). The ALADIN simulations used in the current work can be downloaded from the Med-CORDEX database (www.medcordex.eu).



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



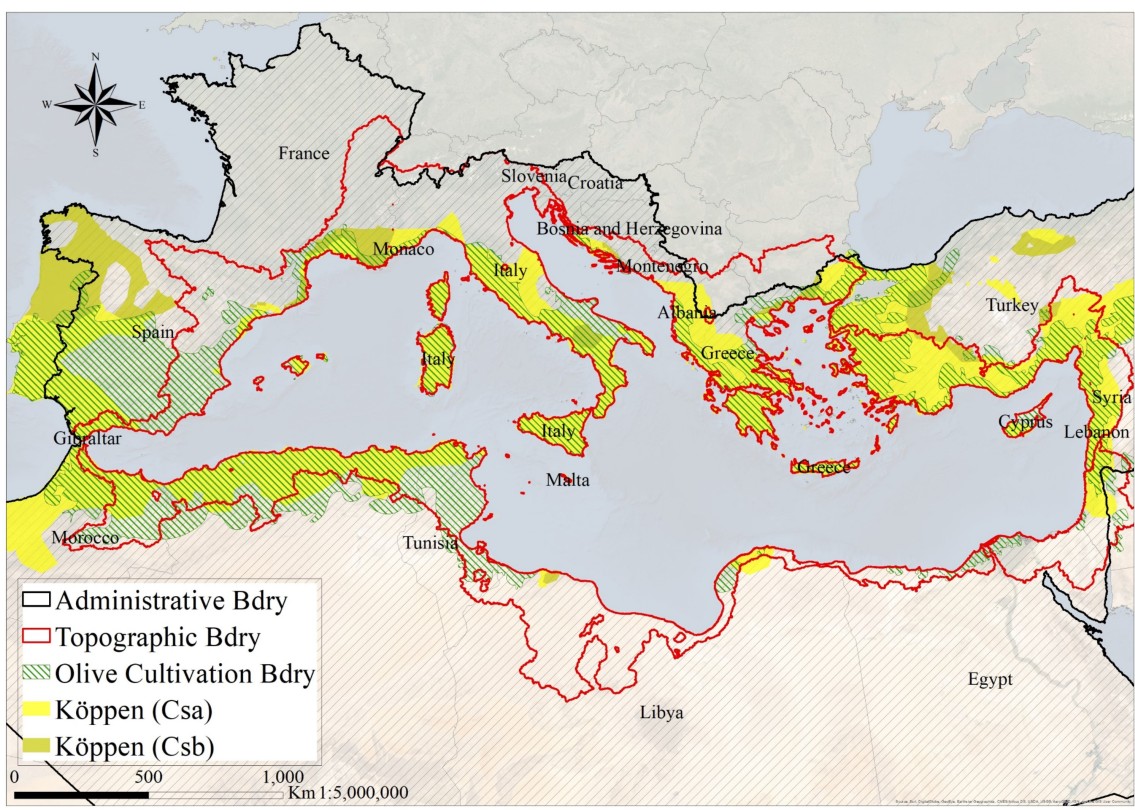

**Figure 1.** Four Mediterranean region boundaries (Bdry) Merheb et al. (2016); first administrative, second topographic Milano (2013), third olive cultivation Moreno (2014) and fourth climatic Peel et al. (2007).

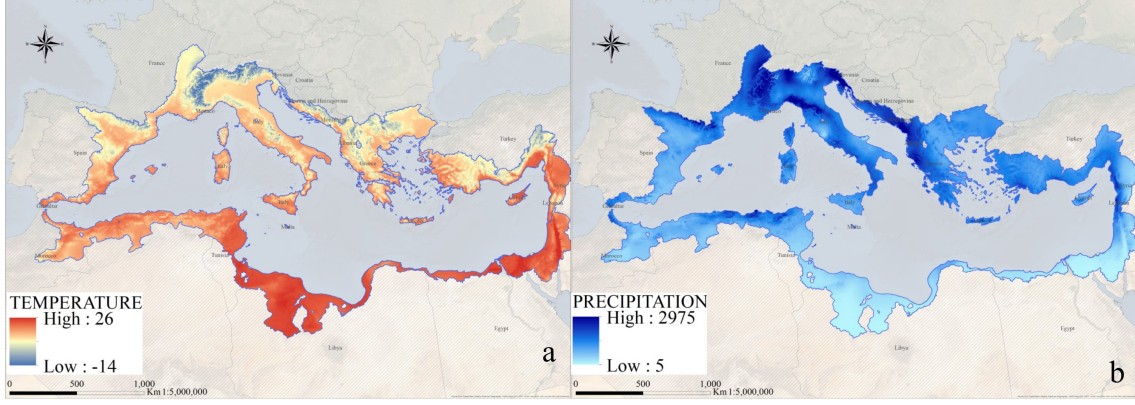

**Figure 2.** WorldClim-2 gridded mean annual temperature in °C (a) and mean annual precipitation in mm (b) from Fick and Hijmans (2017).





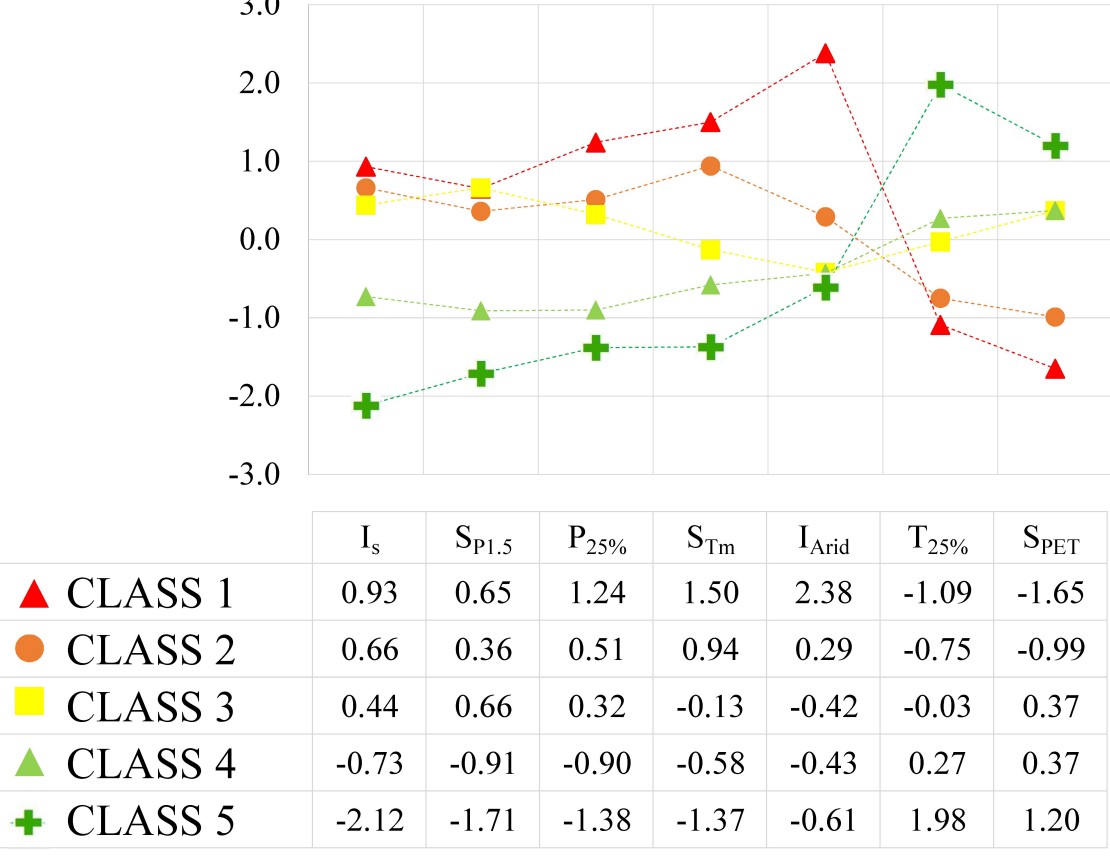

**Figure 3.** Normalized indices values of the five climatic classes kernels from the Mediterranean catchment's classification using WorldClim-2 data.

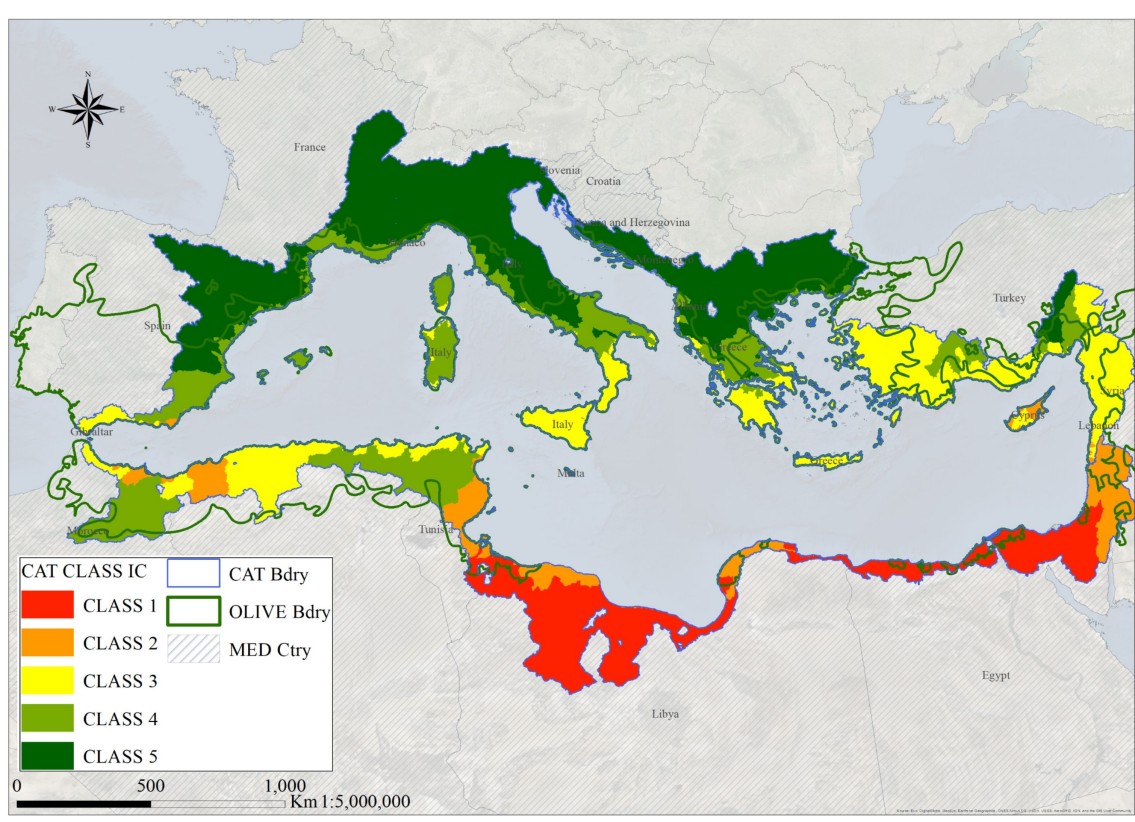

**Figure 4.** Geographical distribution of the Mediterranean climatic classes based on catchments average indices using WorldClim-2 monthly data.

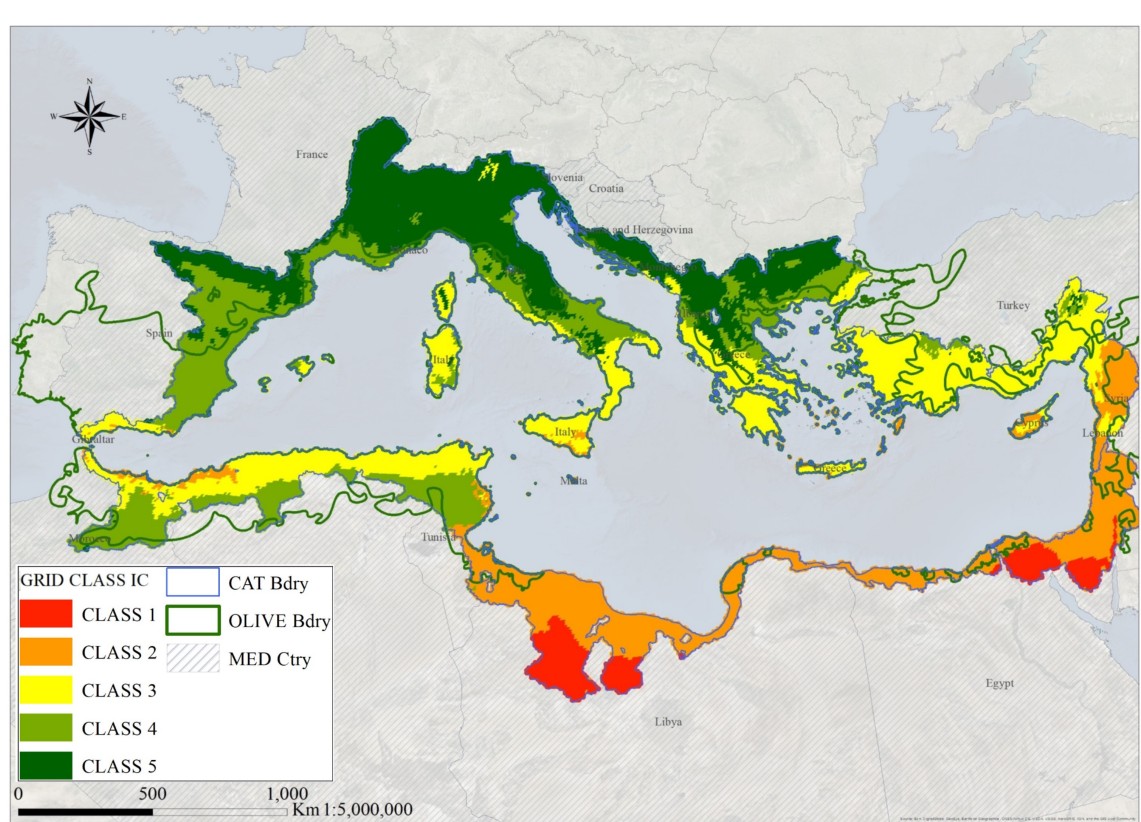

**Figure 5.** Geographical distribution of the Mediterranean climatic classes based on gridded climatic indices using WorldClim-2 monthly data.

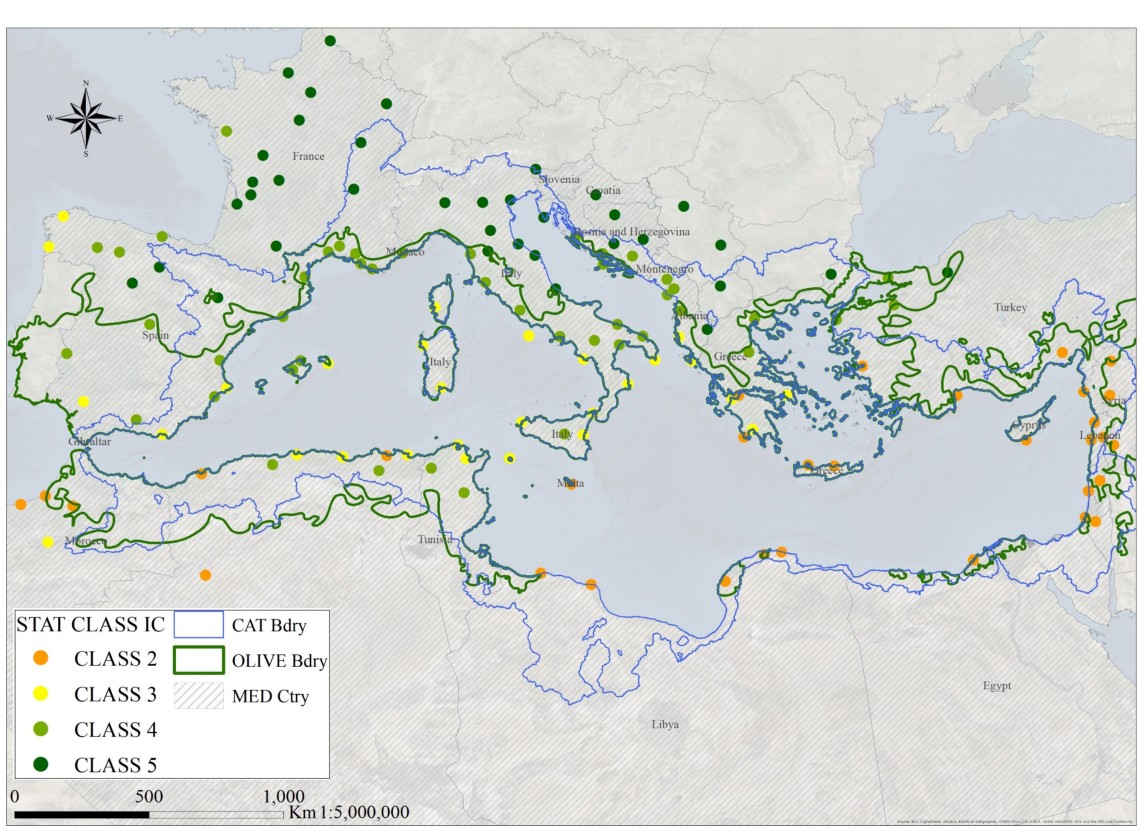

**Figure 6.** Geographical distribution of the Mediterranean climatic classes based on 144 stations climatic indices.




**Figure 7.** Projected geographical distribution of the Mediterranean climatic classes based on WorldClim-2 gridded climatic indices using projected data under RCP 4.5 and 8.5 scenarios for the 2070-2100 period.





**Table 1.** Catchment distribution per area and ratio to total area

| Area range | Number of Catchments | Ratio of Number of Catchments | Total Area $km^2$ | Area Ratio |
|---|---|---|---|---|
| A < 100 $km^2$ | 2333 | 63% | 80,157 | 4% |
| 100 $km^2$ < A < 3000 $km^2$ | 1270 | 35% | 498,614 | 28% |
| A > 3000 $km^2$ | 78 | 2% | 1,202,874 | 68% |

**Table 2.** Statistical summaries for the catchments climatic parameters (Maximum Altitude ($Z_{Max}$), Mean Altitude $Z_{Min}$, Mean Annual Precipitation (MAP), Mean Annual Temperature (MAT), Mean Potential Evapotranspiration (MPET))

|  | Area | $Z_{Max}$ | $Z_{Mean}$ | MAP | MAT | MPET |
|---|---|---|---|---|---|---|
|  | ($km^2$) | (m) | (m) | (mm) | (°C) | (mm) |
| Minimum | 0.01 | -2 | -4 | 39 | 5.1 | 444 |
| Mean | 467.5 | 737 | 255 | 595 | 16.5 | 1136 |
| Maximum | 96619 | 4783 | 1727 | 2004 | 21.7 | 1498 |
| Median | 54.4 | 598 | 185 | 592 | 16.5 | 1127 |



**Table 3.** Climatic Indices Definition

| GROUP | TYPE | CLIMATIC INDICES | DESCRIPTION |
|---|---|---|---|
| | | Seasonality Index $I_s$ | One minus the precipitation ratio between the three most dry and humid consecutive months |
| | | Precipitation Index $P_{25\%}$ and $P_{75\%}$ | Rain value exceeded 25% or 75% of the time |
| I | Climatic Indices based on average monthly rainfall | Peak index $S_{P1.5}, S_{P1.7}, S_{P2}$ | Number of months exceeding the average monthly precipitation by 1.5, 1.7 and 2 times |
| | | Horizontal Inertia Index $I_{Hor}$ | Dispersion of monthly rainfall compared to the annual average |
| | | $\Delta T_1$ | Temperature lag between the coldest and warmest months |
| | | $\Delta T_2$ | Temperature lag between the coldest and warmest three consecutive months |
| II | Climatic Indices based on average monthly temperature | Temperature Index $T_{25\%}$ and $T_{75\%}$ | Temperature value exceeded 25% and 75% of the time |
| | | Peak Index $ST_{1.2}$ | Number of months exceeding the average temperature by 1.2 times |
| | | Degree Day $D_j$ | Decomposition according to the need for habitat heating |
| | | Mean temperature Index $S_{Tm}$ | Number of months exceeding the Mediterranean average temperature Tm 16.4 °C |
| III | Climatic Indices based on precipitation & temperature | Time Lag Index $I_{Decal}$ | Time lag between the coldest and most humid month |
| IV | Climatic index of Evapotranspiration | Aridity Index $I_{Arid}$ | Annual evapotranspiration over annual precipitation $I_{Arid}$ = PET/P |
| | | Threshold Index $S_{PET}$ | Number of months where precipitation exceeds evapotranspiration (PET calculated using Thornthwaite formula) |

**Table 4.** Statistical summaries of the PCA selected climatic indices average for catchments using WorldClim-2 monthly data

| | $I_s$ | $S_{P1.5}$ | $P_{25\%}$ | $S_{Tm}$ | $I_{Arid}$ | $T_{25\%}$ | $S_{PET}$ |
|---|---|---|---|---|---|---|---|
| Minimum | 0.2 | 0.0 | 0.0 | 0.0 | 0.3 | 1.2 | 0.0 |
| Mean | 0.8 | 2.7 | 0.3 | 5.9 | 3.2 | 1.4 | 4.0 |
| Maximum | 1.0 | 5.0 | 0.9 | 10.0 | 38.3 | 2.3 | 12.0 |
| Median | 0.9 | 3.0 | 0.3 | 6.0 | 1.8 | 1.3 | 5.0 |





**Table 5.** Gridded classification decision tree accuracy table. The accuracy rate is calculated in comparison to the K-Means classification

|  | Sample | 1 | 2 | 3 | 4 | 5 | Accuracy |
|---|---|---|---|---|---|---|---|
|  | Class 1 | 636 | 26 | 1 | 1 | 2 | 95.5% |
|  | Class 2 | 86 | 1915 | 131 | 0 | 0 | 89.8% |
| Training | Class 3 | 0 | 118 | 3537 | 186 | 68 | 91.7% |
|  | Class 4 | 1 | 0 | 135 | 2860 | 68 | 93.3% |
|  | Class 5 | 0 | 0 | 1 | 72 | 3511 | 98.0% |
|  | Overall Percentage | 5.4% | 15.5% | 28.6% | 23.4% | 27.0% | 93.6% |
|  | Class 1 | 637 | 33 | 2 | 2 | 0 | 94.5% |
|  | Class 2 | 71 | 1889 | 166 | 0 | 0 | 88.9% |
| Test | Class 3 | 1 | 124 | 3635 | 197 | 11 | 91.6% |
|  | Class 4 | 0 | 0 | 167 | 2912 | 69 | 92.5% |
|  | Class 5 | 0 | 0 | 0 | 83 | 3389 | 97.6% |
|  | Overall Percentage | 5.3% | 15.3% | 29.7% | 23.9% | 25.9% | 93.1% |

**Table 6.** Sample of the decision tree set of rules for the gridded classification (D1, D2, D3, D4 and D5 correspond to distance to kernel of class 1, 2, 3, 4 and 5)

| | |
|---|---|
| CLASS 1 (4 rules) | (D1) < 3.5 and (D2) > 2.2 |
| CLASS 2 (13 RULES) | (D1) < 3.5 and 1.9 < (D2) < 2.2 |
| | 3.5 < (D1) < 4.2 and 2.4 < (D4) < 2.8 and (D2) < 2.2 |
| | 3.5 < (D1) < 4.2 and 2.8 < (D4) < 3.4 |
| | 4.7 < (D1) < 4.8 and (D4) > 3.4 |
| | 4.8 < (D1) < 5.1 and (D4) > 3.4 |
| CLASS 3 (23 rules) | 3.5 < (D1) < 4.2 and 1.8 < (D4) < 2.1 and (D2) < 2.2 |
| | 3.5 < (D1) < 4.2 and 2.1 < (D4) < 2.4 |
| | 5.1 < (D1) < 5.5 and 1.5 < (D4) < 1.8 and (D5) > 1.7 |
| | 5.1 < (D1) < 5.5 and 1.5 < (D4) > 1.8 |
| CLASS 4 (23 rules) | 3.5 < (D1) < 4.2 and (D4) < 1.8 |
| | 3.5 < (D1) < 4.2 and 1.8 < (D4) < 2.1 and (D2) > 2.2 |
| | 5.5 < (D1) < 5.9 and 1.3 < (D5) < 1.7 and 1.2 < (D4) < 1.5 |
| | 5.5 < (D1) < 5.9 and (D5) > 1.7 |
| CLASS 5 (12 rules) | 5.1 < (D1) < 5.5 and 1 < (D4) < 1.2 and (D5) < 1.3 |
| | 5.1 < (D1) < 5.5 and 1.5 < (D4) < 1.5 and (D5) < 1.7 |
| | 5.9 < (D1) < 6.5 and 1.5 < (D4) < 2.4 & (D5) > 1.7 |
| | 5.9 < (D1) < 6.5 and (D4) > 2.4 |
| | (D1) > 6.5 |



**Table 7.** Climatic indices values under RCP scenarios with evolution ratio in italic

|  | Class | A |  | $I_s$ |  | $S_{P1.5}$ |  | $P_{25\%}$ |  | $S_{Tm}$ |  | $I_{Arid}$ |  | $T_{25\%}$ |  | $S_{PET}$ |  |
|---|---|---|---|---|---|---|---|---|---|---|---|---|---|---|---|---|---|
|  | 1 | 5% |  | 0.99 |  | 3.53 |  | 1.70 |  | 9.10 |  | 39.80 |  | 1.33 |  | 0.00 |  |
|  | 2 | 18% |  | 0.98 |  | 3.88 |  | 1.94 |  | 8.76 |  | 9.18 |  | 1.32 |  | 1.00 |  |
| 1970-2000 | 3 | 27% |  | 0.87 |  | 2.90 |  | 1.58 |  | 5.98 |  | 1.75 |  | 1.48 |  | 4.85 |  |
|  | 4 | 22% |  | 0.61 |  | 0.77 |  | 1.29 |  | 5.81 |  | 2.58 |  | 1.47 |  | 3.05 |  |
|  | 5 | 28% |  | 0.41 |  | 0.29 |  | 1.20 |  | 3.66 |  | 0.89 |  | 1.94 |  | 7.56 |  |
|  | 1 | 4% | *0%* | 0.99 | *0%* | 3.45 | *-2%* | 1.79 | *6%* | 9.00 | *-1%* | 39.46 | *-1%* | 1.32 | *-1%* | 0.00 | *0%* |
| RCP 4.5 | 2 | 19% | *1%* | 0.98 | *0%* | 3.65 | *-6%* | 1.99 | *3%* | 8.43 | *-4%* | 9.94 | *8%* | 1.31 | *-1%* | 0.99 | *-1%* |
| 2070-2100 | 3 | 26% | *-1%* | 0.87 | *0%* | 2.90 | *0%* | 1.60 | *1%* | 5.91 | *-1%* | 2.01 | *-15%* | 1.44 | *-3%* | 4.51 | *-7%* |
|  | 4 | 23% | *0%* | 0.66 | *9%* | 1.31 | *70%* | 1.32 | *3%* | 5.64 | *-3%* | 2.73 | *6%* | 1.44 | *-2%* | 2.80 | *-8%* |
|  | 5 | 28% | *0%* | 0.45 | *7%* | 0.50 | *71%* | 1.22 | *2%* | 3.67 | *0%* | 1.06 | *20%* | 1.87 | *-4%* | 7.11 | *-6%* |
|  | 1 | 4% | *0%* | 0.99 | *0%* | 3.44 | *-3%* | 1.83 | *8%* | 8.90 | *-2%* | 38.43 | *-3%* | 1.32 | *-1%* | 0.00 | *0%* |
| RCP 8.5 | 2 | 19% | *1%* | 0.98 | *0%* | 3.60 | *-7%* | 1.98 | *2%* | 8.45 | *-4%* | 10.11 | *10%* | 1.31 | *-1%* | 1.02 | *2%* |
| 2070-2100 | 3 | 26% | *-1%* | 0.86 | *0%* | 2.79 | *-4%* | 1.58 | *0%* | 5.92 | *-1%* | 2.08 | *19%* | 1.44 | *-3%* | 4.38 | *-10%* |
|  | 4 | 24% | *2%* | 0.65 | *7%* | 1.33 | *74%* | 1.34 | *4%* | 5.57 | *-4%* | 2.66 | *3%* | 1.44 | *-2%* | 2.95 | *-3%* |
|  | 5 | 26% | *-2%* | 0.45 | *9%* | 0.57 | *96%* | 1.23 | *3%* | 3.62 | *-1%* | 0.93 | *4%* | 1.89 | *-3%* | 7.32 | *-3%* |