# Peer review of "Mediterranean Specific Climate Classification and Future Evolution Under RCP Scenarios"

_Hydrology and Earth System Sciences, 2019_

## Referee Comment (RC1) · Anonymous Referee #1 · 25 Aug 2019

The manuscript of Allam et al. is a introducing a novel regional climate classification for the Mediterranean region. About the style, the manuscript needs copy-editing for English language check, about the content, despite the scientific relevance of the topic I have two majors concerns that would require substantial modifications to the manuscript:

1- The classification methodology is not robust enough. They authors choose 5 classes without any justification. The authors should provide a robust evaluation of the classification proposed, with the different datasets available. The part about Decision Tree is not sufficiently explained (see specific comments below). The authors should also better highlight the novelty of their approach, by comparison to two recent papers, Barredo et al. 2019 in the reference list, and : Koutroulis A., Dryland changes un-

der different levels of global warming. Science of The Total Environment 655, DOI: 10.1016/j.scitotenv.2018.11.215

2- On the climate change aspect, the use of one single regional climate model simulation is not enough to assess the uncertainties. I suggest either to remove this part or alternatively to strongly upgrade it. The literature review on the topic is very weak and there is a need to include relevant references providing climate scenarios for the whole Mediterranean domain and its different sub-regions. If the authors want to include a climate change study, they could use the ensemble of 50km simulations available in the MedCORDEX experiment. When studying climate change impacts, it is very important to consider the uncertainties from different GCM and RCM simulations, in addition to the uncertainties stemming from the emission scenario.

Specific comments:

Page 1, first lines of introduction: Obviously these sentences are from a text book. Please add the reference.

Page 2 line 11: add reference for MedCORDEX

Page 2, line 15: this part should be moved to a data section later in the text to present the RCM simulations

Page 2, line 25: Ref Tramblay et al 2013 is only for a basin in Morocco. Please add references relevant for the whole Mediterranean.

Page 3, line 1, Rivoire et al 2019 also provided a Mediterranean classification based on P-PET computed from CRU database.

Rivoire, P., Tramblay, Y., Neppel, L., Hertig, E., and Vicente-Serrano, S. M.: Impact of the dry-day definition on Mediterranean extreme dry-spell analysis, Nat. Hazards Earth Syst. Sci., 19, 1629–1638, https://doi.org/10.5194/nhess-19-1629-2019, 2019. Page 3, line 32: "Practiced discipline" = ?

Page 4 line 16: "to be treated in a personal way", strange wording.

Page 4, section 2.2: Not clear what types of catchments are extracted. The authors should precise for which stream orders they extracted the catchment boundaries. Is it for all elementary catchments? Or is there a minimum basin size? For example in the JRC data or HYDROSHED the Pfafstetter coding system is used (de Jager and Vogt, 2010, in the reference list).

Page 4, section 2.3: it would be very useful to also provide a database as an output of this article, a map or GIS layer to have the climatic class for each catchment.

Page 4, section climatic data: The authors should provide a map with the number of station used to build the WordClim database in the Mediterranean and the locations of the 144 weather stations. Several authors have pointed out the strong variability of station density across the Mediterranean region, see:

Zittis G. (2017) Observed rainfall trends and precipitation uncertainty in the vicinity of the Mediterranean, Middle East and North Africa, Theoretical and Applied Climatology. https://doi.org/10.1007/s00704-017-2333-0.

Romera R., Sánchez E., Domínguez M., Gaertner M.Á., Gallardo C. (2015) Evaluation of present-climate precipitation in 25 km resolution regional climate model simulations over Northwest Africa. Clim Res 66(2):125–139.

Raymond, F., Ullmann, A., Camberlin, P., Drobinski, P., and Chateau Smith, C.: Extreme dry spell detection and climatology over the Mediterranean Basin during the wet season, Geophys. Res. Lett., 43, 7196–7204, https://doi.org/10.1002/2016GL069758, 2016.

In addition, the origin of this data is not provided. To which database do they belong? GHCN ?

Page 5, line 1: "5 and 3000" give locations/stations where these values are recorded

Page 5, line 1: Strange that the authors talk about taxonomy for a few lines later explain that it is not useful for climate classifications.

Line 6, line 26: why choose a priori 5 classes? This is a major methodological problem since it is a subjective choice. Usually when performing classifications with kmeans, diagnostic tools such as the Scree plot or Silhouette plot are used to identify and choose the optimal number of clusters. The authors need to clarify and improve this point about the "optimal" number of clusters.

Page 6, section 3.3: What is a Decision Tree? There are no bibliographic references in this section and this is clearly lacking. Do the authors refer to Classification and Regression Trees? (CART, Breiman 1984). How the method is applied is not clear. No need for this type of method to validate a kmeans classification.

Page 7, section 3.4: No presentation of the RCM simulations is provided. In addition, the use of a single simulation is not recommended to provide future scenarios, due to strong differences between different model simulations (Kotlarski et al 2014). This is for sure a weak point in the analysis presented.

See:

Kotlarski, S., Keuler, K., Christensen, O. B., Colette, A., Déqué, M., Gobiet, A., Goergen, K., Jacob, D., Lüthi, D., van Meijgaard, E., Nikulin, G., Schär, C., Teichmann, C., Vautard, R., Warrach-Sagi, K., and Wulfmeyer, V.: Regional climate modeling on European scales: a joint standard evaluation of the EURO-CORDEX RCM ensemble, Geosci. Model Dev., 7, 1297–1333, https://doi.org/10.5194/gmd-7-1297-2014, 2014.

Page 7, section 4.1: It is only in the table 3 that the reader can discover that potential evapotranspiration is computed from the Thornthwaite formula. This formula is most probably not adapted to the Mediterranean context, in particular for climate change scenarios. At least a discussion would be welcome to address this point.

See:

Beguería,S.,Vicente-Serrano,S.M.,Reig,F.,andLatorre,B.:Standardized precipitation evapotranspiration index (SPEI) revisited: parameter fitting, evapotranspiration models, tools, datasets and drought monitoring, Int. J. Climatol., 34, 3001–3023, 2014.

McMahon, T. A., Peel, M. C., Lowe, L., Srikanthan, R., and McVicar, T. R.: Estimating actual, potential, reference crop and pan evaporation using standard meteorological data: a pragmatic synthesis, Hydrol. Earth Syst. Sci., 17, 1331-1363, https://doi.org/10.5194/hess-17-1331-2013, 2013.

Page 8, line 24: "5 classes was the most suitable" this is a contradiction with the methodology described above, where the authors state Line 6, line 26 that they choose 5 classes.

Page 9, line 15: "a similar spatial distribution", similar to what?

Page 9, section 4.4: Usually "validation" refer to the application of a model (or a classification) to data that has not been used for its calibration or training. This is not the case here.

Page 10, section 4.4.3: Again we don't understand what is done here. A validation with a "decision tree"?

Page 10, line 16: "proximity analysis and spatial joint" are not statistical terms but rather obviously Geographic Information Systems (GIS) operations. Please explain clearly which method has been applied.

Page 11, line 3: The reference Colmet-Daage et al 2018 is about the Lez and Aude located in France, and Muga located in northeastern Spain. That is not representative of the whole Mediterranean basin. As mentioned before, the bibliography about climate change projections is rather weak and the authors should cite the relevant literature.

See for example (and the references herein):

Lionello, P., and Scarascia, L.: The relation between climate change in the Mediterranean region and global warming, Reg. Env. Change, 18, 1481-1493, doi: 10.1007/s10113-018-1290-1, 2018.

Wolfgang Cramer, Joël Guiot, Marianela Fader, Joaquim Garrabou, Jean-Pierre Gattuso, Ana Iglesias, Manfred A. Lange, Piero Lionello, Maria Carmen Llasat, Shlomit Paz, Josep Peñuelas, Maria Snoussi, Andrea Toreti, Michael N. Tsimplis, Elena Xoplaki. Climate change and interconnected risks to sustainable development in the Mediterranean. Nature Climate Change, 2018; DOI: 10.1038/s41558-018-0299-2

http://www.medecc.org/climate-and-environmental-change-in-the-mediterranean-main-facts/

Page 11, line 19: It is pretty obvious that "climate is continuous" and should not be mentioned in the conclusions.

Figure 1: Topographic boundaries should be replaced by hydrological boundaries, since what is shown on the map are catchment boundaries.

Figure 4: Green on green is hard to see (for olive boundaries)

Table 6 is very hard to understand

---

## Referee Comment (RC2) · Anonymous Referee #2 · 28 Oct 2019

**General comments**

This article attempts a new classification of the Mediterranean climates using the most recent WorldClim dataset. It also attempts to study how climate change will change the climates of the area of study. This is an interesting object of study, but the article cannot be published without a complete rewriting and improvement of the methods used. Thus, I recommend it to be rejected and resubmitted again in the future.

The main problems of the article are:

1. Incomplete introduction.

2. Insufficient literature review.

3. Messy document structure. Many paragraphs are in the wrong section and/or are incomplete.

4. Methodology insufficiently explained.

5. The article proposes its classification for hydrological purposes, but I do not see any hydrological specificity in the indices and methods used.

6. The catchment based classification seems unnecessary. Its utility should be justified or it should be removed.

7. Some decisions are not well justified (index selection, use of delta change approach, etc.).

8. Insufficient discussion of the results. A Discussion section is necessary.

9. The classification is not sensitive to climate change (the scenarios are very close to the baseline map). Is this a sign that climate change won't have much impact? Or that the method is unable to represent this changes?

**Specific comments**

In my specific comments I will only comment the structural problems. I will not mention the many language issues, such as missing commas or orthography.

Introduction

- Expand the introduction to provide more context and better describe relevant literature.

- P2L13: You should mention that MED-CORDEX is a HyMeX initiative.

- P2L15: Start new paragraph when discussing RCP.

- P2L15-L29: This should be moved to the dataset section, where you describe the Med-CORDEX simulation.

[Figure]

- P2L30-P3L2: Expand. Explain the methodologies that are used, their pros and cons, etc.

- P3L7-L10: Move to datasets section.

- P3L11-L21: Move to Methodology section.

Study area and database

- You describe the different methods to determine the geographical extent of the Mediterranean area, but you do not justify your choice.

- P3L29: Are you sure that "Ecumune" is the right word in this context? To my knowledge Ecumene is the "known world" of the Romans. It seems a more historical term that a geographical one.

- P4L16: "a personal way" is not correct here.

- In "Climatic data" you must justify that WorldClim is appropriate in the Mediterranean. Is the number of stations used by this dataset high enough all over the area?

- P4L31-L32: Move to methodology.

Methodology

- This section is insufficient as it is. You spend more time explaining the history of the methods than describing how you applied them. You should explain all the details so anyone can reproduce your work. You should remove text about the history of the methods and add text detailing you own implementation details. This applies mainly to PCA and K-Means clustering. How do you apply the PCA? How do you normalize the variables? ...

- Some text that should be in this section is found in other sections and vice-versa.

- P5L13: there is no need to talk about taxonomy.

- P5L18-L29: This belongs to the Introduction. You should explain and compare the

methods.

- P5L30-L32: This belongs to the Introduction.

- P6L5-L6: delete the titles of the books. You already provide the citation in the bibliograpy: "... we advise to consult Krzanowski (1988) and Jolliffe (2002).

- P7L6: can you better explain the part about kernel 1? I don't fully understand.

- P7L10: Here you talk about validation. There should be a subsection in Methodology about the validation method.

Section 4

- You should rename this section "Results".

- P7L22-P8L7: you should move this subsection to the Methodology section. Furthermore, you must justify your selection of indices. Why did you choose these indices? Why are they relevant in the Mediterranean? Why are they relevant for hydrology? Did other studies use these same indices?

- P7L22: what do you mean by "subjectively developed"?

- P8L9-L10: You should explain the reduction of indices by means of the correlation matrix in the Methodology section.

- P8L11-14: These results are for catchments or for the grid?

- P8L23: Change subsection title to "4.3. Catchment based classification".

- P8L24: how did you choose 5 classes? Is this arbitrary?

- P8L31: "from the southern tip of Spain to Syria".

- P9L13: Remove 4.4 subsection title.

- P9L14: Change title to "Grid base classification" and change numbering from 4.4.1 to 4.4.

- P9L21-L29: There are many problems here. First, in the data section you don't mention where you obtained the data on the limits of the olive tree geographical domain. Also, I don't see how you conclude that the comparison between the olive domain and your classifications validates the classification. Looking at the maps I don't see that they match well. The olive tree distribution also depends on geology and human practices. I don't see that you can predict the olive tree area using your classification map.

- P9L20: promote the heading one level to 4.5.

- P10L6: How did you divide the data into two subsets. Randomly?

- P10L5: Promote the heading one level to 4.6

- P10L12: Integrate these results into the Results section (4.7).

- P10L13: The delta change approach must be explained in the Methodology. Also, you need to justify that this is the right approach, as you are using percentiles and the delta change approach may produce unrealistic percentiles, such as P75%. Don't forget that the delta change approach only changes the mean of the distribution, but climate change may change the mean and the extremes differently.

- P11L3-L5: I don't understand what you are trying to convey with this paragraph.

- The main result I see in the climate change part is that there is almost no change. This may mean that your method is not sensitive to climate change, or that there is almost no impact of climate change. I guess the right answer is the first one, which means that maybe your classification is not good enough. You should elaborate on that.

A Discussion section is missing before the conclusions.

Conclusions

- Re-write once the other issues have been solved.
Figure 1

- I don't understand the existence of non Mediterranean enclaves within the Mediterranean area. Look at the "islands" found in, for example, Tunisia or Libya.

- Is this Köppen classification made using WorldClim data? You should mention the data source in the datasets section. Same for the Olive Cultivation Boundary.

Figure 4

- I don't find this catchment based classification interesting. Why is interesting to have the whole Rhône, Ebro or Po with the same climate classification when they have diverse climates?

- I cannot see the blue line. What does CAT mean?

- I can't see the green olive domain line.

Figure 5

- I don't understand why the Ebro basin is class 4, as is the coastal area. If you look at the Köppen classification you'll see that the coastal area and the Ebro valley are classified differently, as they have different climates and vegetation types. The same happens with the Rhône and the Po.

- I can't believe that the Alps have the same climate than the Po valley. Your method does not take into account the different climates found at different altitudes.

- I don't understand the yellow area close to Austria, in Italy. You should explain and discuss these missclassifications in the results and discussion sections.

Figure 6

- I see two yellow points in NW Spain. That area has an Köppen classification Cfb. Their climate is very different to the climate of Sicily, for example! I guess this results is due to the fact that you trained your method with data in the Med bassin and, thus, the

method cannot deal well with climates situated outside the domain. If you want to use stations outside the domain, you must train your method with data otside the domain.

---

## Author Comment (AC1) · 25 Nov 2019

**Responses to Referee #1**

Dear Reviewer,

We are very grateful for the constructive comments of Referee #1. We totally agree with all major and specific comments and recommendations. We propose to substantially modify the manuscript in order to respond to all points raised by Reviewer #1 especially by clarifying the presentation of the methodology used, by discussing the choices made for the classification procedure and by adding additional climate change RCM simulations.

In this letter, the comments of Referee #1 are given in black and our response in blue.

Kind regards,

Antoine Allam, Roger Moussa, Wajdi Najem, Claude Bocquillon

**Overview**

The manuscript of Allam et al. is introducing a novel regional climate classification for the Mediterranean region. About the style, the manuscript needs copy-editing for English language check, about the content, despite the scientific relevance of the topic I have two majors concerns that would require substantial modifications to the manuscript:

1- The classification methodology is not robust enough. They authors choose 5 classes without any justification. The authors should provide a robust evaluation of the classification proposed, with the different datasets available. The part about Decision Tree is not sufficiently explained (see specific comments below). The authors should also better highlight the novelty of their approach, by comparison to two recent papers, Barredo et al. 2019 in the reference list, and:

Koutroulis A., Dryland changes under different levels of global warming. Science of The Total Environment 655, DOI: 10.1016/j.scitotenv.2018.11.215

We thank Referee #1 for all his/her comments and agree that the classification methodology should be justified and detailed furthermore.

The methodology will be detailed so anyone could reproduce or apply the study on another database. The methodology will be expanded to include 3 additional subsections in Section 3 (which will have 7 in total):

3.1 Hydrology driven climatic indices

This subsection will be moved from section 4 Results to section 3 Methodology. It will be also expanded to justify the choice of these indices to highlight the hydrological specificity.

3.6 Verification methodology

This subsection will describe the grid-based and ground station classification verification steps within section 3 methodology

3.7 RCP Scenarios

This subsection will detail the procedure to assess the climate change impact on the classification. The delta change calculation and transposition to the WorldClim-2 grid cells using GIS.

Concerning the choice of 5 classes (See response to the comments on "Page 6, line 26"):

We chose 5 classes as we hoped for a classification that delimits the Mediterranean climate from North and South and divides the intermediate coastal zone. Therefore, one class would cover the southern desertic region, another class would cover the northern continental region and 3 classes to cover the intermediate coastal region. Thus, a distribution into 5 classes. A larger number would produce an uninterpretable fragmented classification. A detailed explanation and comparison with subdivisions with more or less 5 classes will be added in the text.

(Koutroulis A. et al., 2018) will be added to highlight the novelty of the work.

A copy-editing for English language check will be done.

2- On the climate change aspect, the use of one single regional climate model simulation is not enough to assess the uncertainties. I suggest either to remove this part or alternatively to strongly upgrade it.

The literature review on the topic is very weak and there is a need to include relevant references providing climate scenarios for the whole Mediterranean domain and its different sub-regions.

If the authors want to include a climate change study, they could use the ensemble of 50km simulations available in the MedCORDEX experiment.

When studying climate change impacts, it is very important to consider the uncertainties from different GCM and RCM simulations, in addition to the uncertainties stemming from the emission scenario.

We also agree that one single simulation is not enough to assess the uncertainties, therefore two additional simulations will be added, CCLM 4-8-19 and RegCM 4.3 for both RCP 4.5 and RCP 8.5 for the 2070-2100 period from the MEDCORDEX project and the discussion section will be expanded accordingly.

All the references mentioned by the referee will be added in their corresponding section, in addition other articles will be added to justify the approach and methodology and discuss the obtained results.

Breiman, L., J. Friedman, C. J. Stone and R. A. Olshen (1984). Classification and Regression Trees, Taylor & Francis.

Cramer Wolfgang, Guiot Joël, Fader Marianela, Joaquim Garrabou, Jean-Pierre Gattuso, Ana Iglesias, Manfred A. Lange, Piero Lionello, Maria Carmen Llasat, Shlomit Paz, Josep Peñuelas, Maria Snoussi, Andrea Toreti, Michael N. Tsimplis, Elena Xoplaki. Climate change and interconnected risks to sustainable development in the Mediterranean. Nature Climate Change, 2018; DOI: 10.1038/s41558-018-0299-2 http://www.medecc.org/climate-and-environmental-change-in-the-mediterraneanmain-facts/

Dell'Aquila, A., A. Mariotti, S. Bastin, S. Calmanti, L. Cavicchia, M. Deque, V. Djurdjevic, M. Dominguez, M. Gaertner and S. Gualdi (2018). "Evaluation of simulated decadal variations over the Euro-Mediterranean region from ENSEMBLES to Med-CORDEX." Climate dynamics 51(3): 857-876.

Diouf, O. C., Weihermüller, L., Ba, K., Faye, S. C., Faye, S., & Vereecken, H. (2016). Estimation of Turc reference evapotranspiration with limited data against the Penman-Monteith Formula in Senegal. Journal of Agriculture and Environment for International Development (JAEID), 110(1), 117-137.

Drobinski, P., N. D. Silva, G. Panthou, S. Bastin, C. Muller, B. Ahrens, M. Borga, D. Conte, G. Fosser, F. Giorgi, I. Güttler, V. Kotroni, L. Li, E. Morin, B. Önol, P. Quintana-Segui, R. Romera and C. Z. Torma (2018). "Scaling precipitation extremes with temperature in the Mediterranean: past climate assessment and projection in anthropogenic scenarios." Climate Dynamics 51(3): 1237-1257.

Giorgi, F. (2006). Regional climate modeling: Status and perspectives. Journal de Physique IV (Proceedings), EDP sciences.

Kotlarski, S., Keuler, K., Christensen, O. B., Colette, A., Déqué, M., Gobiet, A., Goergen, K., Jacob, D., Lüthi, D., van Meijgaard, E., Nikulin, G., Schär, C., Teichmann, C., Vautard, R., Warrach-Sagi, K., and Wulfmeyer, V.: Regional climate modeling on European scales: a joint standard evaluation of the EURO-CORDEX RCM ensemble, Geosci. Model Dev., 7, 1297–1333, https://doi.org/10.5194/gmd-7-1297-2014, 2014.

Koutroulis A., Dryland changes under different levels of global warming. Science of The Total Environment 655, DOI: 10.1016/j.scitotenv.2018.11.215

Lionello, P., and Scarascia, L.: The relation between climate change in the Mediterranean region and global warming, Reg. Env. Change, 18, 1481-1493, doi: 10.1007/s10113-018-1290-1, 2018.

Menne, M. J., I. Durre, B. Korzeniewski, S. McNeal, K. Thomas, X. Yin, S. Anthony, R. Ray, R. S. Vose and B. E. Gleason (2012). "Global historical climatology network-daily (GHCN-Daily), Version 3." NOAA National Climatic Data Center 10: V5D21VHZ.

Raymond, F., Ullmann, A., Camberlin, P., Drobinski, P., and Chateau Smith, C.: Extreme dry spell detection and climatology over the Mediterranean Basin during the wet season, Geophys. Res. Lett., 43, 7196–7204, https://doi.org/10.1002/2016GL069758, 2016.

Rivoire, P., Tramblay, Y., Neppel, L., Hertig, E., and Vicente-Serrano, S. M.: Impact of the dry-day definition on Mediterranean extreme dry-spell analysis, Nat. Hazards Earth Syst. Sci., 19, 1629–1638, https://doi.org/10.5194/nhess-19-1629-2019, 2019.

Romera R., Sánchez E., Domínguez M., Gaertner M.Á., Gallardo C. (2015) Evaluation of present-climate precipitation in 25 km resolution regional climate model simulations over Northwest Africa. Clim Res 66(2):125–139.

Ruti, P. M., S. Somot, F. Giorgi, C. Dubois, E. Flaounas, A. Obermann, A. Dell'Aquila, G. Pisacane, A. Harzallah and E. Lombardi (2016). "MED-CORDEX initiative for Mediterranean climate studies." Bulletin of the American Meteorological Society 97(7): 1187-1208. https://doi.org/10.1175/BAMS-D-14-00176.1

Tramblay, Y. and S. Somot (2018). "Future evolution of extreme precipitation in the Mediterranean." Climatic Change 151(2): 289-302.

Turc, L. (1961). "Estimation of irrigation water requirements, potential evapotranspiration: a simple climatic formula evolved up to date." Ann. Agron 12(1): 13-49.

Zittis, G. (2018). "Observed rainfall trends and precipitation uncertainty in the vicinity of the Mediterranean, Middle East and North Africa." Theoretical and applied climatology 134(3-4): 1207-1230. https://doi.org/10.1007/s00704-017-2333-0.

**Specific comments**

Page 1, first lines of introduction: Obviously these sentences are from a textbook. Please add the reference.

Ok. (Clerget, M., 1937). Les types de temps en Méditerranée. Paper presented at the Annales de géographie.

Page 2 line 11: add reference for MedCORDEX

Ok, reference (Ruti et al., 2016) added at page 2.

Ruti, P. M., S. Somot, F. Giorgi, C. Dubois, E. Flaounas, A. Obermann, A. Dell'Aquila, G. Pisacane, A. Harzallah and E. Lombardi (2016). "MED-CORDEX initiative for Mediterranean climate studies." Bulletin of the American Meteorological Society 97(7): 1187-1208. https://doi.org/10.1175/BAMS-D-14-00176.1

Page 2, line 15: this part should be moved to a data section later in the text to present the RCM simulations

Ok, we agree, it will be moved to section 2.3 climatic data.

Page 2, line 25: Ref Tramblay et al 2013 is only for a basin in Morocco. Please add references relevant for the whole Mediterranean.

Ok. Three references will be added:

Dell'Aquila et al. 2018;

Dell'Aquila, A., A. Mariotti, S. Bastin, S. Calmanti, L. Cavicchia, M. Deque, V. Djurdjevic, M. Dominguez, M. Gaertner and S. Gualdi (2018). "Evaluation of simulated decadal variations over the Euro-Mediterranean region from ENSEMBLES to Med-CORDEX." Climate dynamics 51(3): 857-876.

Drobinski et al., 2018;

Drobinski, P., N. D. Silva, G. Panthou, S. Bastin, C. Muller, B. Ahrens, M. Borga, D. Conte, G. Fosser, F. Giorgi, I. Güttler, V. Kotroni, L. Li, E. Morin, B. Önol, P. Quintana-Segui, R. Romera and C. Z. Torma (2018). "Scaling precipitation extremes with temperature in the Mediterranean: past climate assessment and projection in anthropogenic scenarios." Climate Dynamics 51(3): 1237-1257.

Tramblay et al., 2018

Tramblay, Y. and S. Somot (2018). "Future evolution of extreme precipitation in the Mediterranean." Climatic Change 151(2): 289-302.

Page 3, line 1, Rivoire et al 2019 also provided a Mediterranean classification based on P-PET computed from CRU database.

Rivoire, P., Tramblay, Y., Neppel, L., Hertig, E., and Vicente-Serrano, S. M.: Impact of the dry-day definition on Mediterranean extreme dry-spell analysis, Nat. Hazards Earth Syst. Sci., 19, 1629–1638, https://doi.org/10.5194/nhess-19-1629-2019, 2019.

Ok this reference will be added.

Page 3, line 32: "Practiced discipline"?

Ok. the expression will be modified to "field of practice"

Page 4 line 16: "to be treated in a personal way", strange wording.

Ok, the expression will be removed.

Page 4, section 2.2: Not clear what types of catchments are extracted. The authors should precise for which stream orders they extracted the catchment boundaries. Is it for all elementary catchments? Or is there a minimum basin size? For example, in the JRC data or HYDROSHED the Pfafstetter coding system is used (de Jager and Vogt, 2010, in the reference list).

Ok. the type of catchments will be precised in the text.

We extracted catchments at their outlets to Mediterranean Sea, and only those exceeding 1km$^2$ were considered, hence a total of 3681 catchments all around the Mediterranean.

Page 4, section 2.3: it would be very useful to also provide a database as an output of this article, a map or GIS layer to have the climatic class for each catchment.

Ok. A GIS shapefile of the catchments climatic classes will be provided in supplementary material.

Page 4, section climatic data: The authors should provide a map with the number of stations used to build the WordClim database in the Mediterranean and the locations of the 144 weather stations.

Ok, this map is provided in figures S1 and S2 in the supporting information of Fick and Hijmans (2017) article.

However, we will add this information in the text:

The WorldClim-2 database was built over 23 regions with different coverage for each parameter. For the precipitation an overlap of 3 regions covered the Mediterranean area with a total of 10410 stations for the 3 regions (euw n= 3730; eue n = 3632; naf n = 3048). For average temperature, the Mediterranean was covered by one region (eu1) with number of stations n = 1760; n = 1627 for Maximum temperature and n = 1626 for Minimum temperature;

The location of the 144 stations is shown in figure 6 only, to avoid duplication.

Several authors have pointed out the strong variability of station density across the Mediterranean region, see:

Zittis G. (2017) Observed rainfall trends and precipitation uncertainty in the vicinity of the Mediterranean, Middle East and North Africa, Theoretical and Applied Climatology. https://doi.org/10.1007/s00704-017-2333-0.

Romera R., Sánchez E., Domínguez M., Gaertner M.Á., Gallardo C. (2015) Evaluation of present-climate precipitation in 25 km resolution regional climate model simulations over Northwest Africa. Clim Res 66(2):125–139.

Raymond, F., Ullmann, A., Camberlin, P., Drobinski, P., and Chateau Smith, C.: Extreme dry spell detection and climatology over the Mediterranean Basin during the wet season, Geophys. Res. Lett., 43, 7196–7204, https://doi.org/10.1002/2016GL069758, 2016.

Ok. these references and corresponding information will be added in the following text:

*"The use of ground-based stations time series or gridded observational data is limited by several uncertainties mainly density and interpolation processing methods, especially in Mediterranean region where North African and Levantine countries are poorly covered (Zittis, 2017). Nevertheless,*

*the use of specific indices like seasonality and aridity, which are averaged on 30 years periods and based on monthly and annual values, while avoiding extreme event indices, reduces data quality uncertainties. On the other hand, several studies have revealed the uncertainties connected to the resolution of RCM simulated gridded data in the Mediterranean complex domain (Romera et al., 2015) hence the use high-resolution data like MEDCORDEX 12 km grids and WorldClim-2 1-km and overall, the regional aspect of this study makes it less sensitive to local errors.*

In addition, the origin of this data is not provided. To which database do they belong? GHCN?

These stations belong to Global Historical Climatology Network GHCN (Menne et al., 2012) and recognized by the World Meteorological Organization (WMO), they are available for free access on the portal of the National Administration of Oceans and Atmosphere of the United States (NOAA). The average length of data series is 60 years and range between 30 and 120 years at monthly time step. The 1960 - 1990 period is common to all stations. The data quality was verified (i.e. ellipse of Bois, 1986) and only complete hydrological years were retained for indices calculation. Detailed information will be added in the manuscript.

Page 5, line 1: "5 and 3000" give locations/stations where these values are recorded

Ok. the following information will be added

*"...reflecting the wide variability of mean annual precipitation ranging between 5 ("Jabal el Aswad desert in Libya") and 3000 mm (Kobarid in Slovenia) and mean annual temperature ranging between -14°C (Mont Blanc, Alps, France) and +26°C (Karak, Jordan) where some catchments receive 50 times more than others the amount of precipitation while being 4 times colder."*

Page 5, line 1: Strange that the authors talk about taxonomy for a few lines later explain that it is not useful for climate classifications.

Ok, the paragraph about taxonomy will be removed.

Page 6, line 26: why choose a priori 5 classes? This is a major methodological problem since it is a subjective choice. Usually when performing classifications with kmeans, diagnostic tools such as the Scree plot or Silhouette plot are used to identify and choose the optimal number of clusters. The authors need to clarify and improve this point about the "optimal" number of clusters.

Ok, the following information will be added:

Figure 1 and text in page 5 line 30, shows that the Mediterranean hydrological boundary includes in addition to Csa and Csb climates, the desertic BWh and Bsk and continental Cf and Cs climates.

We chose 5 classes as we hoped for a classification that delimits the Mediterranean climate from North and South and divides the intermediate coastal zone. Therefore, one class would cover the southern desertic region, another class would cover the northern continental region and 3 classes to cover the intermediate coastal region. Thus, a distribution into 5 classes. A larger number would produce an uninterpretable fragmented classification.

Page 6, section 3.3: What is a Decision Tree? There are no bibliographic references in this section and this is clearly lacking. Do the authors refer to Classification and Regression Trees? (CART, Breiman 1984). How the method is applied is not clear. No need for this type of method to validate a k-means classification.

Ok, the following information will be added the subsection 3.4 decision tree

*"A decision tree is a set of distance criteria or questions in the form of hierarchy that leads to an intended classification (Breiman 1984). To classify new points or stations, it suffices to define the distance criterion to the various kernels of the climatic classes by predicting values of a dependent variable based on values of predictor variables from a reference classification."*

It is more a verification of K-means classification rather than a validation.

Page 7, section 3.4: No presentation of the RCM simulations is provided.

We will add a brief description of RCM as presented by Giorgi (2006)

Giorgi, F. (2006). Regional climate modeling: Status and perspectives. Journal de Physique IV (Proceedings), EDP sciences.

*"RCM or Regional Climate Model was introduced in late 1980's as a nested technique into Global Climate Models GCM to consider regional scale climatic forcings caused by the complex physiographic features and small scale circulation features. (Giorgi, 2006). The primary application of RCM has been in the development of climate change scenarios of which we mention ALADIN RCM (Aire Limitée Adaptation dynamique Développement InterNational) developed by Météo France and applied for EURO-CORDEX and MED-CORDEX projects at 12-km spatial resolution (Tramblay et al., 2013)."*

In addition, the use of a single simulation is not recommended to provide future scenarios, due to strong differences between different model simulations (Kotlarski et al 2014). This is for sure a weak point in the analysis presented.

See:

Kotlarski, S., Keuler, K., Christensen, O. B., Colette, A., Déqué, M., Gobiet, A., Goergen, K., Jacob, D., Lüthi, D., van Meijgaard, E., Nikulin, G., Schär, C., Teichmann, C., Vautard, R., Warrach-Sagi, K., and Wulfmeyer, V.: Regional climate modeling on European scales: a joint standard evaluation of the EURO-CORDEX RCM ensemble, Geosci. Model Dev., 7, 1297–1333, https://doi.org/10.5194/gmd-7-1297-2014, 2014.

We agree therefore at least two additional RCM scenarios will be added to the climate change section CCLM 4-8-19 and RegCM 4.3 for both RCP 4.5 and 8.5 scenarios.

Page 7, section 4.1: It is only in the table 3 that the reader can discover that potential evapotranspiration is computed from the Thornthwaite formula. This formula is most probably not adapted to the Mediterranean context, in particular for climate change scenarios. At least a discussion would be welcome to address this point.

See:

Beguería,S.,Vicente-Serrano,S.M.,Reig,F.,andLatorre,B.:Standardized precipitation evapotranspiration index (SPEI) revisited: parameter ïn˘A¸tting, evapotranspiration models, tools, datasets and drought monitoring, Int. J. Climatol., 34, 3001–3023, 2014.

McMahon, T. A., Peel, M. C., Lowe, L., Srikanthan, R., and McVicar, T. R.: Estimating actual, potential, reference crop and pan evaporation using standard meteorological data: a pragmatic synthesis, Hydrol. Earth Syst. Sci., 17, 1331-1363, https://doi.org/10.5194/hess-17-1331-2013, 2013.

Ok, this information will be corrected.

*"The evapotranspiration was estimated according to Turc's formula (Turc, 1961), chosen for its application simplicity and adequacy to Mediterranean areas as it was originally developed for southern France and North African countries (Diouf, 2016). Turc's formula is mainly based on temperature and radiation, two stable parameters on the regional scale which reduces the uncertainties when using regionalized dataset such as WorldClim-2."*

Turc, L. (1961). "Estimation of irrigation water requirements, potential evapotranspiration: a simple climatic formula evolved up to date." Ann. Agron 12(1): 13-49.

Diouf, O. C., Weihermüller, L., Ba, K., Faye, S. C., Faye, S., & Vereecken, H. (2016). Estimation of Turc reference evapotranspiration with limited data against the Penman-Monteith Formula in Senegal. Journal of Agriculture and Environment for International Development (JAEID), 110(1), 117-137.

Page 8, line 24: "5 classes was the most suitable" this is a contradiction with the methodology described above, where the authors state Line 6, line 26 that they choose 5 classes.

Ok. In reference to the answer on the comment of Page 6 line 26, the following information will be added:

*"We chose 5 classes as we hoped for a classification that delimits the Mediterranean climate from North and South and divides the intermediate coastal zone. Therefore, one class would cover the southern desertic region, another class would cover the northern continental region and 3 classes to cover the intermediate coastal region. Thus, a distribution into 5 classes. A larger number would produce an uninterpretable fragmented classification."*

Sentence corrected: "The K-Means classification shows in Figure 4 a distribution into 5 classes where"

Page 9, line 15: "a similar spatial distribution", similar to what?

A spatial distribution similar to the catchment indices classification.

Page 9, section 4.4: Usually "validation" refer to the application of a model (or a classification) to data that has not been used for its calibration or training. This is not the case here.

We agree, it's a verification of the results rather than a validation. It will be modified in the text.

Page 10, section 4.4.3: Again we don't understand what is done here. A validation with a "decision tree"?

Ok. the word "Validation" will be removed, and the following information will be added.

*"We generated a decision tree based on the distances to the clusters' kernels obtained from the gridded indices classification. The aim of this decision tree is to easily reproduce the classification with same kernels rather than repeat the whole classification process which will modify the clusters and their kernels. In this way, the decision tree will permit to follow up the climate evolution and its impact on the classification under other scenarios."*

Page 10, line 16: "proximity analysis and spatial joint" are not statistical terms but rather obviously Geographic Information Systems (GIS) operations. Please explain clearly which method has been applied.

After the calculation of delta changes between baseline period 1970-2000 and projected period 2070-2100, MED-CORDEX grid was overlaid on WorldClim-2 grid and using spatial join algorithm, the attributes of the delta change grid cells were transposed to WorldClim-2 grid cells, to obtain the

projected data on WorldClim-2. The indices were then recalculated using the projected values of monthly temperatures and precipitation.

Spatial join tool corresponds to joining attributes from a source feature to a target source based on the spatial relationship. The source feature is the calculated MED-CORDEX delta change grid and the target feature is the WorldClim-2 grid.

Proximity analysis corresponds to finding target features located within a buffer zone or a distance of a source feature. In our case. The MED-CORDEX grid is the source feature and the WorldClim-2 grid is the target feature. The features of MEDCORDEX grid cells were transposed to the nearest grid cells of WorldClim-2.

Page 11, line 3: The reference Colmet-Daage et al 2018 is about the Lez and Aude located in France, and Muga located in northeastern Spain. That is not representative of the whole Mediterranean basin. As mentioned before, the bibliography about climate change projections is rather weak and the authors should cite the relevant literature.

See for example (and the references herein): Lionello, P., and Scarascia, L.: The relation between climate change in the Mediterranean region and global warming, Reg. Env. Change, 18, 1481-1493, doi: 10.1007/s10113-018-1290-1, 2018.

Wolfgang Cramer, Joël Guiot, Marianela Fader, Joaquim Garrabou, Jean-Pierre Gattuso, Ana Iglesias, Manfred A. Lange, Piero Lionello, Maria Carmen Llasat, Shlomit Paz, Josep Peñuelas, Maria Snoussi, Andrea Toreti, Michael N. Tsimplis, Elena Xoplaki. Climate change and interconnected risks to sustainable development in the Mediterranean. Nature Climate Change, 2018; DOI: 10.1038/s41558-018-0299-2   http://www.medecc.org/climate-and-environmental-change-in-the-mediterraneanmain-facts/

Ok, we agree. The following references Lionello et al., 2018 and Cramer et al., 2018 will be added.

This section will be developed furthermore with to the two additional scenarios results.

Page 11, line 19: It is pretty obvious that "climate is continuous" and should not be mentioned in the conclusions.

Ok. this expression will be removed

Figure 1: Topographic boundaries should be replaced by hydrological boundaries, since what is shown on the map are catchment boundaries.

Ok, "topographic" will be replaced by "hydrological".

[Figure]

Figure 1 Four Mediterranean region boundaries (Bdry) (Merheb et al. 2016); first administrative, second hydrological (Milano 2013), third olive cultivation (Moreno 2014) and fourth climatic (Peel et al. 2007)

Figure 4: Green on green is hard to see (for olive boundaries)

The dark green colour was modified for a brighter green that should be seen clearly. All figures will be modified accordingly.

[Figure]

Figure 4. Geographical distribution of the Mediterranean climatic classes based on catchments average indices using WorldClim-2 monthly data.

Table 6 is very hard to understand

An example of how a rule could be applied was added to the legend. As an example, for class 1, if the distance to kernel 1 (D1) is below 3.5 and the distance to kernel 2 (D2) is above 2.2, then the grid cell belongs to class 1

---

## Author Comment (AC2) · 25 Nov 2019

**Responses to Referee #2**

Dear Reviewer,

We are very grateful for the constructive comments of Referee #2. We totally agree with all major and specific comments and recommendations. We propose to substantially modify the manuscript and totally restructure it according to the referee's advises. We will expand the introduction and methodology as recommended. The bibliography will be expanded.

We give in this letter our responses to the comments in blue.

Kind regards,

Antoine Allam, Roger Moussa, Wajdi Najem, Claude Bocquillon

**Overview**

This article attempts a new classification of the Mediterranean climates using the most recent WorldClim dataset. It also attempts to study how climate change will change the climates of the area of study. This is an interesting object of study, but the article cannot be published without a complete rewriting and improvement of the methods used. Thus, I recommend it to be rejected and resubmitted again in the future.

We thank you for finding the topic interesting. We agree on your comments and the article will be restructured accordingly.

The main problems of the article are:

1. Incomplete introduction.

Ok, we agree. The introduction will be expanded in order to include more details on the background of the climatic classification and its purpose. We will expand and detail the literature on the climatic classification within Mediterranean context.

2. Insufficient literature review.

Ok, we agree, the following references will be added:

Breiman, L., J. Friedman, C. J. Stone and R. A. Olshen (1984). Classification and Regression Trees, Taylor & Francis.

Cramer Wolfgang, Guiot Joël, Fader Marianela, Joaquim Garrabou, Jean-Pierre Gattuso, Ana Iglesias, Manfred A. Lange, Piero Lionello, Maria Carmen Llasat, Shlomit Paz, Josep Peñuelas, Maria Snoussi, Andrea Toreti, Michael N. Tsimplis, Elena Xoplaki. Climate change and interconnected risks to sustainable development in the Mediterranean. Nature Climate Change, 2018; DOI: 10.1038/s41558-018-0299-2 http://www.medecc.org/climate-and-environmental-change-in-the-mediterraneanmain-facts/

Dell'Aquila, A., A. Mariotti, S. Bastin, S. Calmanti, L. Cavicchia, M. Deque, V. Djurdjevic, M. Dominguez, M. Gaertner and S. Gualdi (2018). "Evaluation of simulated decadal variations over the Euro-Mediterranean region from ENSEMBLES to Med-CORDEX." Climate dynamics 51(3): 857-876.

Diouf, O. C., Weihermüller, L., Ba, K., Faye, S. C., Faye, S., & Vereecken, H. (2016). Estimation of Turc reference evapotranspiration with limited data against the Penman-Monteith Formula in Senegal. Journal of Agriculture and Environment for International Development (JAEID), 110(1), 117-137.

Drobinski, P., N. D. Silva, G. Panthou, S. Bastin, C. Muller, B. Ahrens, M. Borga, D. Conte, G. Fosser, F. Giorgi, I. Güttler, V. Kotroni, L. Li, E. Morin, B. Önol, P. Quintana-Segui, R. Romera and C. Z. Torma (2018). "Scaling precipitation extremes with temperature in the Mediterranean: past climate assessment and projection in anthropogenic scenarios." Climate Dynamics 51(3): 1237-1257.

Giorgi, F. (2006). Regional climate modeling: Status and perspectives. Journal de Physique IV (Proceedings), EDP sciences.

Kotlarski, S., Keuler, K., Christensen, O. B., Colette, A., Déqué, M., Gobiet, A., Goergen, K., Jacob, D., Lüthi, D., van Meijgaard, E., Nikulin, G., Schär, C., Teichmann, C., Vautard, R., Warrach-Sagi, K., and Wulfmeyer, V.: Regional climate modeling on European scales: a joint standard evaluation of the EURO-CORDEX RCM ensemble, Geosci. Model Dev., 7, 1297–1333, https://doi.org/10.5194/gmd-7-1297-2014, 2014.

Koutroulis A., Dryland changes under different levels of global warming. Science of The Total Environment 655, DOI: 10.1016/j.scitotenv.2018.11.215

Lionello, P., and Scarascia, L.: The relation between climate change in the Mediterranean region and global warming, Reg. Env. Change, 18, 1481-1493, doi: 10.1007/s10113-018-1290-1, 2018.

Menne, M. J., I. Durre, B. Korzeniewski, S. McNeal, K. Thomas, X. Yin, S. Anthony, R. Ray, R. S. Vose and B. E. Gleason (2012). "Global historical climatology network-daily (GHCN-Daily), Version 3." NOAA National Climatic Data Center 10: V5D21VHZ.

Raymond, F., Ullmann, A., Camberlin, P., Drobinski, P., and Chateau Smith, C.: Extreme dry spell detection and climatology over the Mediterranean Basin during the wet season, Geophys. Res. Lett., 43, 7196–7204, https://doi.org/10.1002/2016GL069758, 2016.

Rivoire, P., Tramblay, Y., Neppel, L., Hertig, E., and Vicente-Serrano, S. M.: Impact of the dry-day definition on Mediterranean extreme dry-spell analysis, Nat. Hazards Earth Syst. Sci., 19, 1629–1638, https://doi.org/10.5194/nhess-19-1629-2019, 2019.

Romera R., Sánchez E., Domínguez M., Gaertner M.Á., Gallardo C. (2015) Evaluation of present-climate precipitation in 25 km resolution regional climate model simulations over Northwest Africa. Clim Res 66(2):125–139.

Ruti, P. M., S. Somot, F. Giorgi, C. Dubois, E. Flaounas, A. Obermann, A. Dell'Aquila, G. Pisacane, A. Harzallah and E. Lombardi (2016). "MED-CORDEX initiative for Mediterranean climate studies." Bulletin of the American Meteorological Society 97(7): 1187-1208. https://doi.org/10.1175/BAMS-D-14-00176.1

Tramblay, Y. and S. Somot (2018). "Future evolution of extreme precipitation in the Mediterranean." Climatic Change 151(2): 289-302.

Turc, L. (1961). "Estimation of irrigation water requirements, potential evapotranspiration: a simple climatic formula evolved up to date." Ann. Agron 12(1): 13-49.

Zittis, G. (2018). "Observed rainfall trends and precipitation uncertainty in the vicinity of the Mediterranean, Middle East and North Africa." Theoretical and applied climatology 134(3-4): 1207-1230. https://doi.org/10.1007/s00704-017-2333-0.

3. Messy document structure. Many paragraphs are in the wrong section and/or are incomplete.

Ok. The articles will be restructured according to the reviewer's comments. The introduction and methodology will be expanded. A discussion section will be added.

4. Methodology insufficiently explained.

The methodology will be detailed so anyone could reproduce or apply the study on another database. The methodology will be expanded to include 3 additional subsections in Section 3 (which will have 7 in total):

3.1 Hydrology driven climatic indices

This subsection will be moved from section 4 Results to section 3 Methodology. It will be also expanded to justify the choice of these indices to highlight the hydrological specificity.

3.6 Verification methodology

This subsection will describe the grid-based and ground station classification verification steps within section 3 methodology

3.7 RCP Scenarios

This subsection will detail the procedure to assess the climate change impact on the classification. The delta change calculation and transposition to the WorldClim-2 grid cells using GIS.

5. The article proposes its classification for hydrological purposes, but I do not see any hydrological specificity in the indices and methods used.

This paper described the first stage of a larger study within the thesis of Allam on Mediterranean hydrology entitled "Hydrological characterization of Mediterranean catchments based on climatic and physiographic features"

Most of the climatic indices considered in this study reflects the Mediterranean seasonality and precipitation intermittence such as the frequency indicators ($I_S$, $P_{25\%}$, $P_{75\%}$) or periodical indicator like ($S_{P1.5}$, $S_{P1.7}$, $S_{P2}$) or the dispersion index of precipitation $I_{Hor}$.

The same could be said about temperature indices ($\Delta T1$, $\Delta T2$, $T_{25\%}$ & $T_{75\%}$ $S_{T1.2}$) which describe the seasonality and variability of evapotranspiration or intermittence of dry and humid seasons.

All these indices reflect the precipitation seasonality and variability which constitute the main climatic forcing into the hydrological regimes of Mediterranean rivers.

Motivated by the quest for a specific hydrology for the Mediterranean we first established a new high-resolution climatic classification for hydrology purposes based on Mediterranean specific climate indices like precipitation seasonality and aridity that play an important role in the hydrological mechanisms of Mediterranean catchments and flow intermittence.

This classification is useful with the projection of future scenarios, in following up hydrological (water resources management, floods, droughts, etc.) and ecohydrological applications such as Mediterranean agriculture like olive cultivation and other environmental practices.

Hydrologically, Haines in 1988 has classified river regimes based on monthly average flows only and identified the Mediterranean regimes under 3 of the 15 global classes and found a clear relation to Köppen's Mediterranean climate (Haines et al., 1988).

The hydrological similarity between catchments should be judged based on specific metrics taking into account the complexity of the environmental factors impacting the catchment response (Wagener, Sivapalan, & Troch, 2007). This approach has been adopted in Mediterranean regional studies like Oueslati in 2015 who classified Mediterranean rivers into 6 flow regimes, based on Richter's hydrological indices and broad-scale catchment characteristics (Richter et al., 1996; Oueslati et al., 2015) and by Di Prinzio for the classification of 300 Italian catchments making use of Self Organization Maps (Di Prinzio et al., 2011).

Di Prinzio, M., Castellarin, A., & Toth, E. (2011). Data-driven catchment classification: application to the pub problem. Hydrol. Earth Syst. Sci., 15(6), 1921-1935. doi:10.5194/hess-15-1921-2011

Haines, A., Finlayson, B., & McMahon, T. (1988). A global classification of river regimes. *Applied Geography, 8*(4), 255-272.

Oueslati, O., De Girolamo, A. M., Abouabdillah, A., Kjeldsen, T. R., & Lo Porto, A. (2015). Classifying the flow regimes of Mediterranean streams using multivariate analysis. Hydrological Processes, 29(22), 4666-4682. doi:10.1002/hyp.10530

Richter, B. D., Baumgartner, J. V., Powell, J., & Braun, D. P. (1996). A Method for Assessing Hydrologic Alteration within Ecosystems. Conservation biology, 10(4), 1163-1174. doi:doi:10.1046/j.1523-1739.1996.10041163.x

Wagener, T., Sivapalan, M., & Troch, P. (2007). Catchment Classification and Hydrologic Similarity. Geography Compass, 901-931.

6. The catchment-based classification seems unnecessary. Its utility should be justified, or it should be removed.

This section will be clarified in the text.

The catchments constitute the elementary unit of any hydrological study, and as clarified in our response to the previous question this classification is for hydrological purposes. By classifying the Mediterranean catchments climatically and physiographically, we will be able to classify them hydrologically and predict the hydrological characteristics on Ungauged basins.

7. Some decisions are not well justified (index selection, use of delta change approach, etc.).

Ok we agree, the sections concerning the indices choice and delta change approach will be detailed and justified in the text.

The delta change consists of the temperature and precipitation difference between the baseline period 1970-2000 and the projected period 2070-2100.

After the calculation of delta changes, MED-CORDEX grid was overlaid on WorldClim-2 grid and using spatial join algorithm, the attributes of the delta change grid cells were transposed to WorldClim-2 grid cells, to obtain the projected data on WorldClim-2. The indices were then recalculated using the projected values of monthly temperatures and precipitation.

Spatial join tool corresponds to joining attributes from a source feature to a target source based on the spatial relationship. The source feature is the calculated MED-CORDEX delta change grid and the target feature is the WorldClim-2 grid.

Proximity analysis correspond to finding target features located within a buffer zone or a distance of a source feature. In our case. The MED-CORDEX grid is the source feature and the WorldClim-2 grid is the target feature. The features of MEDCORDEX grid cells were transposed to the nearest grid cells of WorldClim-2.

8. Insufficient discussion of the results. A Discussion section is necessary.

Ok we agree, a discussion section will be added to include the new results of the additional simulations.

9. The classification is not sensitive to climate change (the scenarios are very close to the baseline map). Is this a sign that climate change won't have much impact? Or that the method is unable to represent these changes?

Ok, we agree that one simulation is not enough to highlight the climate change impact on the classification, therefore two RCM simulations CCLM 4-8-19 and RegCM 4.3 for both RCP 4.5 and 8.5 scenarios will be added. The discussion will be expanded accordingly.

However, and according to the obtained results, we can see that the classes area and spatial evolution is not major however, the seasonality index $I_S$ has varied between 7% and 9% as for the $S_{P1.5}$ it increased between 70% and 96% for classes 4 and 5.

**Specific comments**

In my specific comments I will only comment the structural problems. I will not mention the many language issues, such as missing commas or orthography.

Ok, the text will be reviewed by a colleague whose English is his native language.

**Introduction**

- Expand the introduction to provide more context and better describe relevant literature.

Ok, we agree, the introduction will be expanded to include more literature on the climatic classification within Mediterranean context.

- P2L13: You should mention that MED-CORDEX is a HyMeX initiative.

Ok, this information will be added, and the sentence will become:

"Med-CORDEX, a HyMeX initiative, (Ruti et al., 2016) is part of the…"

- P2L15: Start new paragraph when discussing RCP.

Ok.

- P2L15-L29: This should be moved to the dataset section, where you describe the Med-CORDEX simulation.

Ok, this paragraph will be moved to 2.3 Climatic data, under MEDCORDEX paragraph (3).

- P2L30-P3L2: Expand. Explain the methodologies that are used, their pros and cons, etc.

Text will be expanded and used methodologies of each mentioned study will be included.

- P3L7-L10: Move to datasets section.

Ok, paragraph was moved to section 2.3 Climatic data.

- P3L11-L21: Move to Methodology section. Study area and database

Ok, paragraph was moved as an introduction to section 3. Methodology.

- You describe the different methods to determine the geographical extent of the Mediterranean area, but you do not justify your choice.

Ok, we agree. As the main purpose is hydrological, we chose the hydrological boundary as a study area. The hydrological boundary could also be considered as the topographic, morphological or landform boundary which constitute an important boundary to microclimates.

- P3L29: Are you sure that "Ecumune" is the right word in this context? To my knowledge Ecumene is the "known world" of the Romans. It seems a more historical term that a geographical one.

It was also used to define a geographical extent of the old inhabited world.

- P4L16: "a personal way" is not correct here.

Ok, the expression will be modified to be:

"*Since the geographic extent of the study is very wide, the delimitation of catchments was imported from international references.*"

- In "Climatic data" you must justify that WorldClim is appropriate in the Mediterranean. Is the number of stations used by this dataset high enough all over the area?

Ok, this section will be modified to include the details on the number of stations considered in WorldClim-2 for the Mediterranean region. Refer to figures S1 and S2 in the supporting information of Fick and Hijmans (2017) article.

"*The WorldClim-2 database was built over 23 regions with different coverage for each parameter. For the precipitation an overlap of 3 regions covered the Mediterranean area with a total of 10410 stations for the 3 regions (euw n= 3730; eue n = 3632; naf n = 3048). For average temperature, the Mediterranean was covered by one region (eu1) with number of stations n = 1760; n = 1627 for Maximum temperature and n = 1626 for Minimum temperature;*"

- P4L31-L32: Move to methodology.

The sentence "Both classifications were compared for validation." will be moved from 2.3 climatic data, (1) Woldclim to 3.4 Adopted Methodology to become:

"*The classification was verified and compared to WorldClim-2 gridded indices and ground stations indices*"

**Methodology**

- This section is insufficient as it is. You spend more time explaining the history of the methods than describing how you applied them. You should explain all the details so anyone can reproduce your work. You should remove text about the history of the methods and add text detailing you own implementation details. This applies mainly to PCA and K-Means clustering. How do you apply the PCA? How do you normalize the variables? ...

Ok, we agree. The historical part will be reduced. The PCA implementation steps will be detailed so the methodology could be reproduced.

- Some text that should be in this section is found in other sections and vice-versa.

Ok we agree, the article structure will be modified according to the reviewer's comments with additional modifications of the author.

- P5L13: there is no need to talk about taxonomy.

Ok, the sentence about taxonomy was removed.

- P5L18-L29: This belongs to the Introduction. You should explain and compare the methods.

Ok, this section will be moved to the introduction. And the different methods will be explained.

*Genetic classification* is based on the cause of the climate. A genetic system relies on information about climate elements like solar radiation, air masses, pressure systems, etc.

*Bioclimatology* studies the relationship between the climate and the distribution of living beings and their communities on the Earth. This discipline began to take shape on connecting mean climate values (temperature and precipitation) with values on areas occupied by plants and plant formation.

*Bioclimatic classification* provides a typology of bioclimates which has an accurate relationship between vegetational models and climate values and considering the high predictive value of bioclimatic units it could be used in the studies of biodiversity and habitats.

*Agro-climatic classification* describes the inter relationship between agronomy, farming systems and climate

- P5L30-L32: This belongs to the Introduction.

Ok, this section will be moved to the introduction.

- P6L5-L6: delete the titles of the books. You already provide the citation in the bibliography: "... we advise to consult Krzanowski (1988) and Jolliffe (2002).

Ok, it will be modified.

- P7L6: can you better explain the part about kernel 1? I don't fully understand.

This information will be added to the text.

*"By forcing the kernel 1, corresponding to class 1, into the first node of the decision tree, the classification rules (distances to kernels) were generated according to kernel 1. Otherwise, we could have forced kernel 2 (of class 2) into the first node of decisions and obtained another set of rules."*

- P7L10: Here you talk about validation. There should be a subsection in Methodology about the validation method.

Ok, a new subsection will be added 3.5 Validation methodology.

**Section 4**

- You should rename this section "Results".

Ok, the section title will be modified to "results".

- P7L22-P8L7: you should move this subsection to the Methodology section. Furthermore, you must justify your selection of indices. Why did you choose these indices? Why are they relevant in the Mediterranean? Why are they relevant for hydrology? Did other studies use these same indices?

Ok, this subsection will be moved to the methodology, and choice of indices will be justified as follow:

Most of the climatic indices considered in this study reflects the Mediterranean seasonality and precipitation intermittence such as the frequency indicators ($I_S$, $P_{25\%}$, $P_{75\%}$) or periodical indicator like ($S_{P1.5}$, $S_{P1.7}$, $S_{P2}$) or the dispersion index of precipitation $I_{Hor}$.

The same could be said about temperature indices ($\Delta T1$, $\Delta T2$, $T_{25\%}$ & $T_{75\%}$ $S_{T1.2}$) which describe the seasonality and variability of evapotranspiration or intermittence of dry and humid seasons.

All these indices reflect the precipitation seasonality and variability which constitute the main climatic forcing into the hydrological regimes of Mediterranean rivers.

- P7L22: what do you mean by "subjectively developed"?

The indices were chosen subjectively to emphasize Mediterranean climate contribution into hydrological behaviour of Mediterranean catchments. Sentence will be modified as follow:

*"The hydrology driven independent climatic indices were chosen subjectively and developed at the catchment scale from WorldClim-2 monthly average data."*

- P8L9-L10: You should explain the reduction of indices by means of the correlation matrix in the Methodology section.

Ok, the following sentence will be added to the methodology section.

*"The number of indices is reduced at two steps. The first step is based on the correlation matrix, where strongly correlated indices higher than 0.85 were eliminated. The second is based on PCA results where indices that doesn't contribute into the principal component that represent the greatest variabilities are eliminated."*

- P8L11-14: These results are for catchments or for the grid?

The results in this section are for catchments as no PCA was applied on grid indices but instead the classification was transposed using the decision tree.

- P8L23: Change subsection title to "4.3. Catchment based classification".

Ok, it will be modified.

- P8L24: how did you choose 5 classes? Is this arbitrary?

Ok, the choice will be justified and the following information will be added:

*"Figure 1 and text in page 5 line 30, shows that the Mediterranean hydrological boundary includes in addition to Csa and Csb climates, the desertic BWh and Bsk and continental Cf and Cs climates. We chose 5 classes as we hoped for a classification that delimits the Mediterranean climate from North and South and divides the intermediate coastal zone. Therefore, one class would cover the southern desertic region, another class would cover the northern continental region and 3 classes to cover the intermediate coastal region. Thus, a distribution into 5 classes. A larger number would produce an uninterpretable fragmented classification."*

- P8L31: "from the southern tip of Spain to Syria".

Ok, the text will be modified.

- P9L13: Remove 4.4 subsection title.

Ok, it will be modified.

- P9L14: Change title to "Grid base classification" and change numbering from 4.4.1 to 4.4.

Ok, it will be modified.

- P9L21-L29: There are many problems here. First, in the data section you don't mention where you obtained the data on the limits of the olive tree geographical domain. Also, I don't see how you conclude that the comparison between the olive domain and your classifications validates the classification. Looking at the maps I don't see that they match well. The olive tree distribution also depends on geology and human practices. I don't see that you can predict the olive tree area using your classification map.

Ok, we agree. The comparison between the olive cultivation boundary and climatic classification needs more clarification.

The olive cultivation boundary was included in section 2.1 under the description of the Agricultural-bioclimatic boundary (P4L9) and it was obtained from Moreno, 2014.

We agree that olive tree cultivation depends on factors other than climatic, however in this paper we were just pointing out on the relationship between the typical Mediterranean climate expressed by classes 2, 3, and 4 and the olive cultivation which matches as olive tree cultivation boundary and classes 1 and 5 boundaries rarely intersect.

- P9L20: promote the heading one level to 4.5.

Ok, it will be modified.

- P10L5: Promote the heading one level to 4.6

Ok, it will be modified.

- P10L6: How did you divide the data into two subsets. Randomly?

Yes randomly. The sentence will be modified to "The total population of gridded indices was divided randomly into two equal subsets".

- P10L12: Integrate these results into the Results section (4.7).

Ok, it will be modified.

- P10L13: The delta change approach must be explained in the Methodology. Also, you need to justify that this is the right approach, as you are using percentiles and the delta change approach may produce unrealistic percentiles, such as P75%. Don't forget that the delta change approach only changes the mean of the distribution, but climate change may change the mean and the extremes differently.

Ok, this section will be moved to methodology into a new subsection 3.7 RCP Scenarios.

- P11L3-L5: I don't understand what you are trying to convey with this paragraph.

The following paragraph about indices and data uncertainties will be added into the subsection 2.3 climatic data.

 The use of ground-based stations time series or gridded observational data is limited by several uncertainties mainly density and interpolation processing methods, especially in Mediterranean region where North African and Levantine countries are poorly covered (Zittis, 2017). Nevertheless, the use of specific indices like seasonality and aridity, which are averaged on 30 years periods and based on monthly and annual values, while avoiding extreme event indices, reduces data quality uncertainties. On the other hand, several studies have revealed the uncertainties connected to the resolution of RCM simulated gridded data in the Mediterranean complex domain (Romera et al., 2015) hence the use high-resolution data like MEDCORDEX 12 km grids and WorldClim-2 1-km and overall, the regional aspect of this study makes it less sensitive to local errors.

- The main result I see in the climate change part is that there is almost no change. This may mean that your method is not sensitive to climate change, or that there is almost no impact of climate change. I guess the right answer is the first one, which means that maybe your classification is not good enough. You should elaborate on that.

We agree, therefore two additional scenarios (CCLM 4-8-19 and RegCM 4.3) will be included in the climate change section for RCP 4.5 and RCP 8.5. The classification evolution under all scenarios will be discussed accordingly.

A Discussion section is missing before the conclusions.

A discussion section will be added

**Conclusions**

- Re-write once the other issues have been solved.

Ok, we agree

Figure 1

- I don't understand the existence of non Mediterranean enclaves within the Mediterranean area. Look at the "islands" found in, for example, Tunisia or Libya.

Actually, these are enclosed catchment that drain into a lake or pond and not into the Mediterranean Sea.

- Is this Köppen classification made using WorldClim data? You should mention the data source in the datasets section. Same for the Olive Cultivation Boundary.

The references are mentioned in the dataset section.

The Köppen classification was revised by Peel in 2007 based on a set of stations and not based on WorldClim-2 data.

Figure 4

- I don't find this catchment based classification interesting. Why is interesting to have the whole Rhône, Ebro or Po with the same climate classification when they have diverse climates?

Yes, Rhône, Ebro and Po are very wide catchments (above than 10000 km$^2$) and have diverse climates, however smaller catchments (less than 3000 km$^2$) belong to only one climatic class. It is interesting to

have a catchment based climatic classification that shall be cross analysed with a catchment based physiographic classification (article in preparation) that both classifications will be used for a hydrological characterization of Mediterranean catchments. Hence the utility of hydrology oriented climatic indices.

- I cannot see the blue line. What does CAT mean?

CAT = Catchment boundary, thin blue line that indicated the hydrological boundary, legend modified to hydrological boundary

- I can't see the green olive domain line.

The dark green colour was modified for a brighter green that should be seen clearly

[Figure]

Figure 4. Geographical distribution of the Mediterranean climatic classes based on catchments average indices using WorldClim-2 monthly data.

Figure 5

- I don't understand why the Ebro basin is class 4, as is the coastal area. If you look at the Köppen classification you'll see that the coastal area and the Ebro valley are classified differently, as they have different climates and vegetation types. The same happens with the Rhône and the Po.

The climatic classification might differ for wide catchments classification between the catchment based classification and grid based classification because of the climatic variables averaging for catchments.

- I can't believe that the Alps have the same climate than the Po valley. Your method does not take into account the different climates found at different altitudes.

Alps and Po valley don't have the same climate, but they belong to the same climatic class according to this classification approach. They both share close seasonality index (around 0.47) and close aridity index $I_{Arid}$ (around 1.06). It should be reminded that this classification was developed for hydrological

purposes within the Mediterranean context, which makes it different from Köppen's or other pure climatic classifications.

- I don't understand the yellow area close to Austria, in Italy. You should explain and discuss these missclassifications in the results and discussion sections.

The classification gives a deterministic result where each grid should belong to one of the 5 classes. Grid classification is a result of a set of decision rules based on its distances to classes kernels where distance is calculated from the grid indices each with a different range of values. Therefore, grids could belong to different classes but were classified according to the nearest kernel.

Figure 6

- I see two yellow points in NW Spain. That area has an Köppen classification Cfb. Their climate is very different to the climate of Sicily, for example! I guess this results is due to the fact that you trained your method with data in the Med bassin and, thus, the method cannot deal well with climates situated outside the domain. If you want to use stations outside the domain, you must train your method with data outside the domain.

Not all the stations located within Csa or Csb and in the Mediterranean region belong to the same climatic class.

The two yellow points stations are closer to the class 3 kernel than of other classes' kernels. They might have neighbouring indices' values as of other Class 3 points without sharing the same climate.